# Fatty-acid-induced FABP5/HIF-1 reprograms lipid metabolism and enhances the proliferation of liver cancer cells

Jieun Seo [1,2], Do-Won Jeong [1,2], Jong-Wan Park [1,2,3], Kwang-Woong Lee[4], Junji Fukuda[5] & Yang-Sook Chun[1,2,3✉]

Hypoxia-inducible factor-1 alpha (HIF-1α) is a transcription factor essential for cancer cell survival. The reprogramming of lipid metabolism has emerged as a hallmark of cancer, yet the relevance of HIF-1α to this process remains elusive. In this study, we profile HIF-1α-interacting proteins using proteomics analysis and identify fatty acid-binding protein 5 (FABP5) as a critical HIF-1α-binding partner. In hepatocellular carcinoma (HCC) tissues, both FABP5 and HIF-1α are upregulated, and their expression levels are associated with poor prognosis. FABP5 enhances HIF-1α activity by promoting HIF-1α synthesis while disrupting FIH/HIF-1α interaction at the same time. Oleic-acid treatment activates the FABP5/HIF-1α axis, thereby promoting lipid accumulation and cell proliferation in HCC cells. Our results indicate that fatty-acid-induced FABP5 upregulation drives HCC progression through HIF-1-driven lipid metabolism reprogramming.

[1] Department of Biomedical Sciences, Seoul National University College of Medicine, Seoul 03080, South Korea. [2] Department of Physiology, Seoul National University College of Medicine, Seoul 03080, South Korea. [3] Ischemic/Hypoxic Disease Institute, Seoul National University College of Medicine, Seoul 03080, South Korea. [4] Department of Hepatobiliary and Pancreatic Surgery, Seoul National University Hospital, Seoul 03080, South Korea. [5] Faculty of Engineering, Yokohama National University, Yokohama 240-8501, Japan. ✉email: chunys@snu.ac.kr

The chemical reactions that occur in cells and contribute to energy production or the synthesis of cellular material are referred to as metabolism. Cancer cells initially have limited access to nutrients due to poor vascularization. To overcome this, cancer cells undergo metabolic reprogramming, which is crucial for subsequent rapid tumor growth[1–3]. Apart from the fact that lipids are essential components of plasma membranes, they are also important signaling molecules and energy source; thus, the reprogramming of lipid metabolism has emerged as one of the new hallmarks of cancer[4,5]. Among other changes that occur during lipid-metabolism-reprogramming in cancer, malignant cancer cells increase de novo lipogenesis and exogenous fatty acids uptake and lipid accumulation, while beta-oxidation decreases; these changes drive cancer cell survival, despite the unfavorable conditions in the cancer microenvironment. Therefore, interfering with lipid metabolism in cancer has become a promising therapeutic strategy. However, the precise molecular mechanisms underlying the lipid-metabolism-reprogramming in cancer remain elusive[6–11].

Hypoxia-inducible factor-1 alpha (HIF-1α) is a transcription factor essential for cancer cell survival, as it drives the expression of metabolism-related and survival-related genes in response to low oxygen levels. The role of HIF-1 in glucose metabolism (Warburg effect) has been extensively studied over the last two decades[12–14]. The relevance of HIF-1α in lipid metabolism reprogramming is under-studied, although evidence supporting its role in this process exists. During hypoxia, HIF-1α promotes fatty acid uptake through induction of FABPs (FABP3, FABP7, and FABP4) along with PPARγ, and lipid storage by modulating ADRP, AGPAT2, and LIPIN1 expressions. Furthermore, HIF-1 suppresses fatty acid oxidation through the inhibition of the PGC-1α, CPT1A, and acyl-CoA dehydrogenases and lipolysis through the suppression of ATGL. By doing so, HIF-1α-dependent alteration of lipid metabolism enables cancer cells to promote survival and growth[7,8,15–23]. However, the role of fatty acids in the regulation of HIF-1 activity and cancer cell survival has not been investigated.

In this study, we analyzed the relationship between changes in lipid metabolism and HIF-1α-interacting proteins using proteomics and identified fatty-acid-binding protein 5 (FABP5). FABP5 has been shown to be a cytosolic transporter for oleic acid (OA)[24,25]. It has also been reported that treatment with OA induces FABP5 expression in normal prostate cells and pancreatic islet cells. Furthermore, FABP5 silencing in human brain endothelial cells resulted in a reduction in OA uptake. The importance of FABP5 in metabolic responses has also been highlighted recently, as FABP4/5 knockout mice exhibited a lower incidence of diet-induced obesity and type 2 diabetes[26–28]. In addition to the role of FABP5 in metabolic diseases, FABP5 may also play a critical role in cancer progression. FABP5 is overexpressed in several types of cancer and associated with poor prognosis[29–34]. Considering the previously reported links between lipid metabolism and tumor progression, we hypothesized that FABP5/HIF-1α plays a crucial role in cancer development. Herein, we report that FABP5 and HIF-1α were upregulated in hepatocellular carcinoma (HCC) tissues and their expression levels were associated with poor prognosis. FABP5 induced HIF-1α expression and activity by inhibiting FIH binding and enhancing p300 binding. Under high OA-condition, FABP5/HIF-1α axis was activated and accelerated lipid storage through the induction of lipid-storage-related genes. Additionally, our results indicated the OA/FABP5/HIF-1α axis facilitated HCC cell proliferation, indicating this axis is a promising therapeutic target for reversing cancer-related lipid metabolism reprogramming.

## Results

### The HIF-1α-interacting protein, FABP5, is highly expressed in human HCC tissues and positively correlates with HIF-1α expression.
To identify HIF-1α-interacting proteins that regulate lipid metabolism, we used an anti-HA antibody to pull down the HA-tagged HIF-1α N-terminal fragment, and sequentially analyzed co-purified proteins using liquid chromatography-tandem mass spectrometry. Proteins specifically interacting with the HIF-1α N-terminal fragment were identified by excluding proteins pulled down in the control group (Fig. 1a and Supplementary Fig. 1a). Among these proteins, we focused on FABP5, which is known to regulate lipid metabolism. The direct interaction between HIF-1α and FABP5 was further confirmed via immunoprecipitation after ectopic expression of HA-HIF-1α and Flag/Streptavidin Binding Peptide-FABP5 (F/S-FABP5, Fig. 1b). To investigate the role of FABP5 in HCC progression, FABP5 mRNA levels were analyzed using the NCBI GEO database (dataset GSE41804). FABP5 mRNA expression was elevated in liver tissues from HCC patients compared with tissues from healthy individuals (Fig. 1c). We further confirmed the relationship between FABP5 and HIF-1α target genes using gene set enrichment analysis (GSEA) in FABP5-high and FABP5-low samples (Supplementary Fig. 1b). We found that HIF-1α target genes were enriched in FABP5-high samples (Fig. 1d). We then evaluated the FABP5 and HIF-1α protein levels in HCC tissues obtained from patients. Immunohistochemical analyses indicated that the protein levels of FABP5 and HIF-1α in HCC tissues were higher than those in normal liver tissues. Subsequently, we examined their expression in paired normal and tumor tissues and found that the protein levels of FABP5 and HIF-1α were elevated in tumor tissues compared with adjacent normal tissue (Fig. 1e and Supplementary Fig. 1c). Pearson correlation analysis also revealed a positive correlation between FABP5 and HIF-1α protein levels in human HCC tissues (Fig. 1f). HCC tissues were further divided into FABP5-high or HIF-1α-high and FABP5-low or HIF-1α-low tissues; we found that elevated FABP5 and HIF-1α expression in the tumor was associated with poor tumor-free survival in HCC patients (Fig. 1g). Collectively, these data indicate that HIF-1α interacts with FABP5, and elevated FABP5 and HIF-1α may be involved in HCC progression.

### FABP5 induces HIF-1α upregulation at the translational level.
Having demonstrated the positive correlation between expression levels of HIF-1α and FABP5 in human HCC samples, we then assessed the molecular mechanism underlying FABP5-mediated HIF-1α induction. The ectopic expression of FABP5 resulted in increased nuclear HIF-1α protein levels during both normoxia and hypoxia (Fig. 2a); however, HIF-1α mRNA levels did not change significantly (Fig. 2b and Supplementary Fig. 2a). Moreover, HIF-1α stability was not affected by the ectopic expression of FABP5 (Fig. 2c and Supplementary Fig. 2b). We also examined the de novo synthesis rate of HIF-1α using the proteasome inhibitor MG132 and found that it increased upon ectopic expression of FABP5 (Fig. 2d). HIF-1α translation is regulated by 5′ cap-dependent and IRES-dependent mechanisms. Using a HIF1A 5′-UTR-luciferase reporter system, we investigated the cap-dependent translation of HIF-1α and found that the ectopic expression of FABP5 increased the rate of translation, both during normoxia and hypoxia (Fig. 2e). As the PI3K–AKT–mTOR pathway has been shown to enhance the 5′ cap-dependent translation of HIF-1α, we examined the potential involvement of PI3K–AKT–mTOR signaling in the FABP5-induced HIF-1α upregulation. AKT–mTOR pathway inhibition with MK2206 or rapamycin did not entirely suppress HIF-1α expression, and the phosphorylation levels of AKT or mTOR did

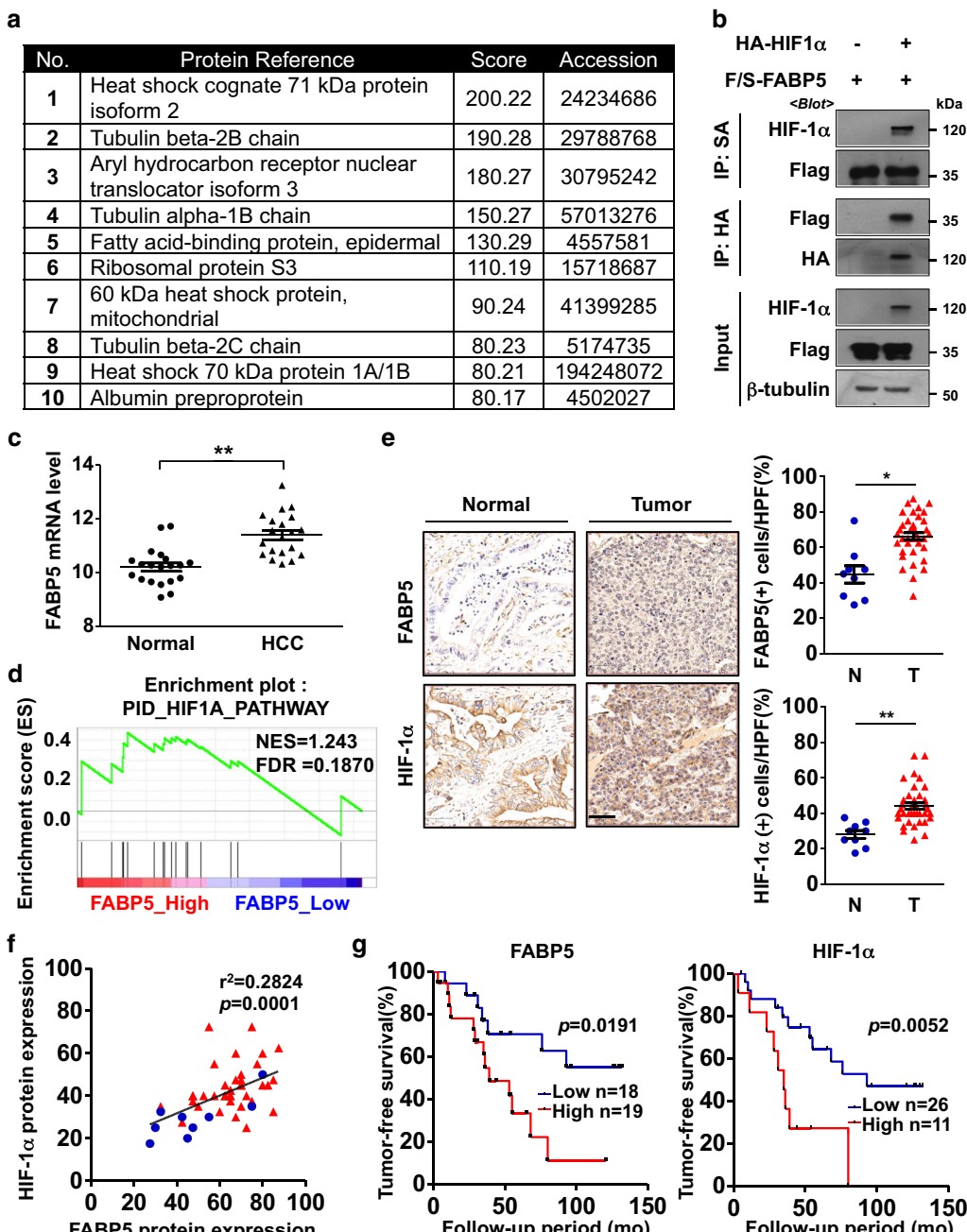

**Fig. 1 FABP5 is highly expressed in human HCC tissues, and its levels positively correlate to HIF-1α levels. a** The top-ranked HIF-1α-interacting proteins, as identified using liquid chromatography-tandem mass spectrometry. 293T cells were transfected with HA-HIF-1α (N-terminal fragment) or HA-tag DNA for control. Cell lysates were collected and purified with antiHA. Filtering was performed based on score >60. **b** 293T cells were transfected with Flag/SBP-FABP5 (F/S-FABP5) with or without HA-HIF-1α and then treated with 10 μM MG132 for 8 h. Cell lysates were subjected to immunoprecipitation using streptavidin-affinity beads or HA-affinity beads and then assessed via western blotting. **c** FABP5 mRNA expression in noncancerous tissues (Normal, $n = 20$) and hepatocellular carcinoma tissues (HCC, $n = 20$), which were obtained from the GEO database (GSE41804). The horizontal lines in all dot plots represent the means ± standard error of the mean (SEM). **$P < 0.0001$ **d** The gene set enrichment analysis (GSEA) plot for the PID_HIF1A_PATHWAY gene set in the FABP5-high and FABP5-low expression groups. **e** Immunohistochemical analysis of FABP5 and HIF-1α protein expression in a human HCC tissue microarray. Representative images are shown on the left, and dot plots of the immunostaining scores are shown in the middle. Dot plots of paired normal adjacent-to-tumor tissues and tumor tissues are shown on the right. The horizontal lines in all dot plots represent the means ± SEM. *$P < 0.05$; **$P < 0.0001$. The scale bar represents 50 μm. **f** Pearson correlation plot showing a positive correlation between FABP5 and HIF-1α protein expression levels in the human tissue microarray. Blue, normal tissue; red, tumor tissue. The *R*-value indicates the Pearson correlation coefficient value. **g** Kaplan–Meier analysis showing the association between FABP5 or HIF-1α expression levels and the tumor-free survival rates in human HCC patients. The high and low groups were determined according to average protein expression levels. The *P*-value was calculated using the log-rank test.

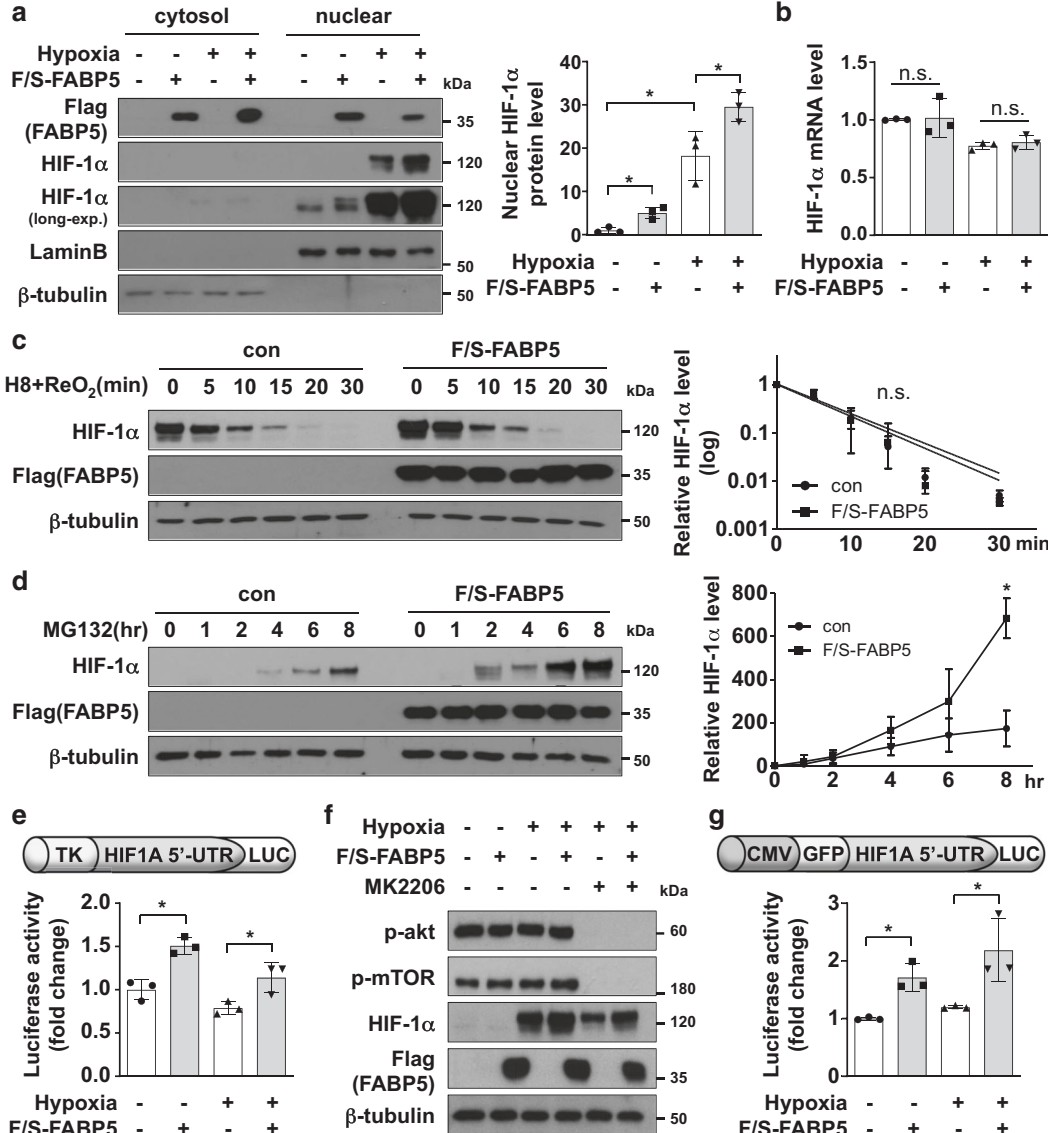

**Fig. 2 FABP5 enhances protein synthesis of HIF-1α via translational upregulation. a** F/S-FABP5-transfected 293T cells were incubated under normoxic or hypoxic conditions for 8 h and then subjected to nuclear extraction. Cytosolic and nuclear fractions were analyzed using western blotting and signal quantification was performed based on Lamin B expression levels (mean ± SD, n = 3; *, P < 0.05). **b** F/S-FABP5-transfected 293T cells were incubated under normoxic or hypoxic conditions for 8 h and then lysed for RNA extraction. HIF-1α mRNA levels were measured using quantitative reverse-transcription polymerase chain reaction (RT-qPCR). Data are represented as the means ± standard deviation (n = 3). n.s. no statistically significant difference between the indicated groups. **c** 293T cells were transfected with control vector or F/S-FABP5 and incubated under hypoxic conditions for 8 h, followed by reoxygenation at 21% O2 for the indicated times. Samples were immunoblotted for HIF-1α, and signal quantification was performed using Image J. HIF-1α band intensities were normalized to the corresponding β-tubulin intensities. The linear regression was plotted, and each symbol represents the mean ± SD (n = 3). *P < 0.05 **d** The transfected 293T cells were incubated with 10 μM MG132 for the indicated times. Western blot intensities were analyzed using Image J, and the band intensities were normalized to the corresponding β-tubulin intensities. Each symbol represents the mean ± SD (n = 3). *P < 0.05. **e, g** 293T cells were co-transfected with the TK-HIF1A-5′-UTR-luciferase plasmid (**e**) or CMV-GFP-HIF1A-5′-UTR-luciferase plasmid (**g**), the CMV-B-galactosidase plasmid, and F/S-FABP5. After incubation for 24 h, the transfected cells were incubated under normoxic or hypoxic conditions for 16 h. Luciferase activity was measured and normalized to the respective β-galactosidase activity. Each bar represents the mean, and error bars represent SD (n = 3). *P < 0.05 **f** Transfected 293T cells were incubated under normoxic or hypoxic conditions and treated with 1 μM MK2206 for 8 h. Cell lysates were subjected to immunoblotting with the indicated antibodies and the blots were quantified by using Image J (see also Supplementary Fig. 2c).

not increase upon ectopic expression of FABP5 (Fig. 2f and Supplementary Fig. 2c, d). We also investigated the IRES-dependent HIF-1α translation using a CMV-GFP-HIF1A 5′-UTR-luciferase reporter system. Luciferase activity was increased by the ectopic expression of FABP5, both during normoxia and hypoxia (Fig. 2g). Taken together, our results revealed that FABP5 induces HIF-1α upregulation at the protein level by enhancing its 5′ cap-dependent and IRES-dependent translation.

**FABP5 promotes HIF-1α transcriptional activity by enhancing HIF-1α and p300 interaction and by inhibiting the interaction between HIF-1α and FIH in the cytosol**. We then explored HIF-1-dependent transcriptional activity by employing a luciferase reporter plasmid containing a hypoxia-response element (HRE) from the erythropoietin enhancer region and confirmed that luciferase activity increased in response to FABP5 overexpression under normoxic and hypoxic conditions; however, EPO vector

containing the mutated HRE did not increase luciferase activity under hypoxia (Fig. 3a). Furthermore, we evaluated the endogenous mRNA levels of the HIF-1 target genes BNIP3L and VEGF using quantitative reverse-transcription polymerase chain reaction (PCR), and found that the mRNA levels of both genes

were elevated in FABP5-overexpressing cells (Fig. 3b). Co-immunoprecipitation was performed to verify the domain of HIF-1α that binds to FABP5, and we confirmed that FABP5 binds directly to the N-terminal domain of HIF-1α (Fig. 3c). Because the C-terminal transactivation domain (CAD) is responsible for

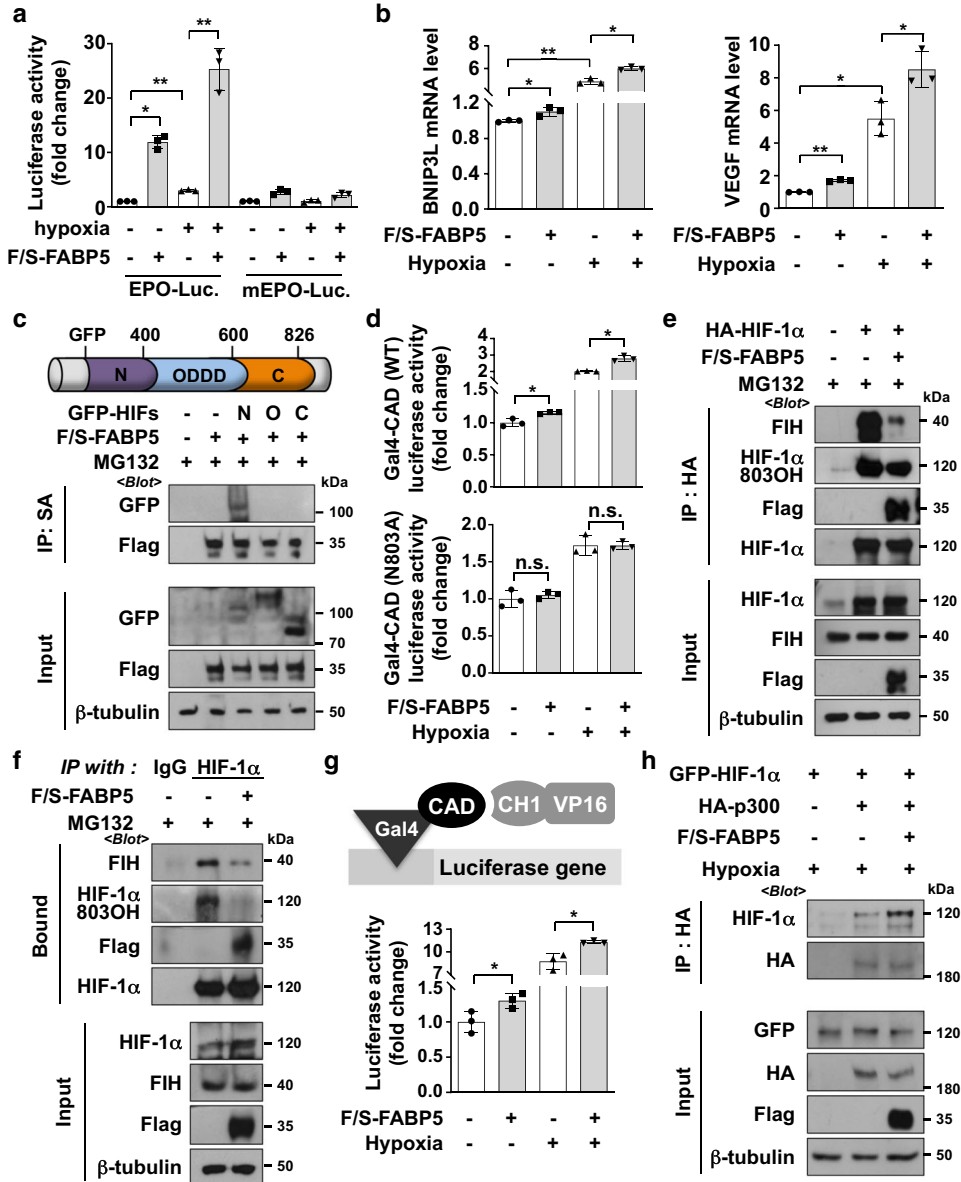

**Fig. 3 FABP5 reinforces the p300-dependent transcriptional activity of HIF-1α by inhibiting the interaction between HIF-1α and FIH. a** 293T cells were co-transfected with the HRE- or mutated HRE-luciferase plasmid, CMV-β-galactosidase, and F/S-FABP5, and incubated under the indicated conditions for 16 h. Luciferase activity (mean ± SD, $n = 3$) was normalized to the respective β-galactosidase activity. *$P < 0.05$; **$P < 0.0001$. **b** 293T cells, which had been transfected with F/S-FABP5, were incubated under the indicated conditions for 16 h and lysed for RNA extraction. BNIP3L and VEGF mRNA levels were measured using RT-qPCR. Each bar represents the mean, and error bars represent SD ($n = 3$); *$P < 0.05$; **$P < 0.0001$. **c** 293T cells were transfected with the F/S-FABP5 with GFP-HIF-1α-N-terminal (aa: 1–400) or GFP-HIF-1α-ODDD (aa: 300–600) or GFP-HIF-1α-C-terminal (aa: 600–826), respectively. After incubation with 10 μM MG132 for 8 h, cells were subjected to immunoprecipitation using streptavidin-affinity beads. The bound proteins were subjected to immunoblotting. (aa: amino acids). **d** 293T cells were co-transfected with Gal4-promoter-Luc reporter vector, Gal4-HIF-1α-CAD (or CAD N803A), and the F/S-FABP5 vector. Cells were incubated under normoxic or hypoxic conditions for 16 h and then lysed for a luciferase assay. Data are presented as the means ± SD ($n = 3$). *$P < 0.05$; n.s., not significant. **e**, 293T cells were co-transfected with HA-HIF-1α and F/S-FABP5 and incubated with 10 μM MG132 for 8 h. Cells were harvested and subjected to immunoprecipitation using HA-affinity beads, and the bound proteins were immunoblotted with the indicated antibodies. **f** F/S-FABP5 transfected 293T cells were incubated with 10 μM MG132 for 8 h. Cell lysates were immunoprecipitated using IgG or HIF-1α antibodies, and the bound proteins were immunoblotted. **g** 293T cells, which had been transfected with F/S-FABP5, Gal4-Luc reporter, Gal4-CAD, and VP16-CH1 vectors, were incubated under the indicated conditions for 16 h. Cell extracts were subjected to a luciferase assay. Data are presented as the means ± SD ($n = 3$). *, $P < 0.05$. **h** GFP-HIF-1α, HA-p300, and F/S-FABP5 were overexpressed in 293T cells, and cells were incubated under hypoxic conditions for 8 h. Lysed proteins were precipitated using HA-affinity beads and subjected to western blotting.

HIF-1α activation, we tested the activity of HIF-1α-CAD using a Gal4 reporter system. As the Gal4-CAD fusion protein is constantly expressed regardless of oxygen levels, this reporter system could indicate the HIF-1α transcriptional activity irrespective of the HIF-1α protein level. The activity of HIF-1α-CAD was enhanced by the ectopic expression of FABP5, both during normoxia and hypoxia. As CAD activity is inhibited by FIH, which hydroxylates the N803 residue of CAD, we compared the effect of FABP5 ectopic expression on both wild-type Gal4-CAD and N803A-mutant Gal4-CAD. Interestingly, FABP5 could not reinforce CAD activity in N803A-mutant Gal4-CAD (Fig. 3d). Furthermore, the extent of the interaction between FIH and HIF-1α was decreased by FABP5 overexpression, and the hydroxylation levels of HIF-1α-N803 also decreased markedly (Fig. 3e, f). Next, we confirmed the interaction between p300 and HIF-1α using a mammalian two-hybrid system with Gal4-HIF-1α-CAD and p300-CH1-VP16. The CAD-CH1 interaction was enhanced by FABP5 overexpression under normoxic or hypoxic conditions (Fig. 3g). Furthermore, the ectopic expression of FABP5 enhanced the interaction between HIF-1α and p300 (Fig. 3h). Taken together, these data strongly suggest that FABP5 inhibits HIF-1α hydroxylation by interfering with FIH binding and enhances p300-dependent HIF-1 transcriptional activity.

**OA-mediated FABP5 induction promotes HIF-1α activity in HCC cells**. Next, we assessed the relevance of the FABP5/HIF-1α axis to HCC cells. Treatment of HepG2 cells with OA resulted in an increase in the endogenous levels of FABP5 protein and mRNA in a dose-dependent manner (Fig. 4a, b). Furthermore, OA treatment increased the levels of HIF-1α protein but not those of mRNA (Fig. 4b–d). We also confirmed that OA-mediated HIF-1α upregulation at the protein level is FABP5-dependent using cells in which FABP5 was silenced (Fig. 4e). Next, we performed immunocytochemistry with MG132-treated HepG2 cells, and found that endogenous FABP5 co-exists with HIF-1α in the cytoplasm (Fig. 4f). We then explored FABP5 silencing effect on the interaction between HIF-1α and FIH. The result revealed that OA-induced FABP5 attenuates the HIF-1α-FIH binding; however FABP5 knockdown recovered the interaction between HIF-1α and FIH, and the hydroxylation levels of HIF-1α N803 (Fig. 4g). Finally, we assessed whether HIF-mediated transcription was enhanced by OA. Using an EPO-luciferase reporter vector, we found that the enhanced HIF-1 activity was reversed in FABP5 knockdown cells, both under normoxic and hypoxic conditions (Fig. 4h). These results verified that OA induces FABP5, thereby increasing the de novo synthesis of HIF-1α at the translational level and activating its transcriptional activity by inhibiting FIH-dependent hydroxylation and promoting p300 binding (Fig. 4i).

**The FABP5–HIF-1α axis facilitates lipid accumulation in HCC cells**. We then examined the lipid-mediated regulation of FABP5 under an excessive influx of OA. First, we quantified the lipid accumulation in cells, as one of the main functions of hepatocytes is fat storage in the form of lipid droplets. Lipid droplets accumulated after OA treatment of HepG2 cells were stained with Nile red (Supplementary Fig. 3a). FABP5 or HIF-1α silencing by siRNA transfection reduced the number of lipid droplets under hypoxic conditions compared to numbers observed in control cells (Fig. 5a and Supplementary Fig. 3b). We also assessed the mRNA levels of genes involved in lipid storage, β-oxidation, and lipolysis, as these genes are involved in the regulation of free fatty-acid storage in the form of cellular lipid droplets. Genes involved in lipid storage, including ACSL1, GPAT, LIPIN1, and DGAT2, were upregulated after OA treatment, even under hypoxic

conditions; however, their expressions were repressed by FABP5 or HIF-1α silencing. Nevertheless, the expression of the lipid-β-oxidation-related gene CPT1A and lipolysis-related gene ATGL was induced by FABP5 or HIF-1α knockdown after OA treatment and under hypoxic conditions (Fig. 5b). Because ACSL1 plays a key role in converting long-chain fatty acids into fatty acyl-CoA, we performed chromatin immunoprecipitation coupled quantitative PCR to examine the direct binding of HIF-1α to the ACSL1 promoter. Among three possible HREs, the proximal region (P3 in Fig. 5c) was identified as the OA-induced HIF-1α-binding site. We then assessed whether FABP5 promotes HIF-1α binding to the ACSL1 promoter. The ability of OA to promote HIF-1α binding to the HRE in the ACSL1 promoter was attenuated in cells in which FABP5 was silenced (Fig. 5c). GSEA analysis confirmed the enrichment of fatty-acid-metabolism-related gene sets based on FABP5 expression levels. We also verified that β-oxidation-related gene sets were expressed at higher levels in FABP5/HIF-1α-low samples (Fig. 5d). Furthermore, the expression levels of genes involved in lipid droplet formation were evaluated in human HCC tissues. ACSL1, GPAT, LIPIN1, and DGAT2 mRNA levels were upregulated in HCC tissues (Fig. 5e). Taken together, the OA-induced FABP5/HIF-1α pathway drives the expression of genes involved in lipid accumulation, thus promoting fatty-acid storage in HCC cells in the form of lipid droplets.

**OA facilitates HepG2 cell survival through the FABP5–HIF-1α axis**. Lipid droplets serve as a primary source of cell membrane components in rapidly proliferating cells, and correlate with poor prognosis in several types of cancer. Given that the FABP5/HIF-1α axis stimulates lipid-droplet formation after OA exposure and under hypoxic conditions, we further evaluated the potential of OA to enhance cell survival through the activation of the FABP5/HIF-1α axis. Interestingly, liver tissues from HCC patients expressed FABP5 and HIF-1α target genes at high levels (Fig. 1c, d), and high levels of FABP5 and HIF-1α were associated with the expression of cell-cycle-pathway gene signatures (Fig. 6a and Supplementary Fig. 4a). Furthermore, cell numbers decreased when FABP5 or HIF-1α was silenced under normoxic and hypoxic conditions (Supplementary Fig. 4b). The results of a colony-formation assay revealed that OA treatment increased colony number and size, and these phenomena were suppressed when FABP5 or HIF-1α was silenced (Fig. 6b). To further assess the role of the FABP5/HIF-1α axis in tumor spheroid growth, we used an in vitro three-dimensional (3D) cell-culture system with si-FABP5 or si-HIF-1α-treated HepG2 cells. Spheroids were formed on day 1, and HepG2 cells were treated with OA and harvested on day 5 (Fig. 6c and Supplementary Fig. 4c). Interestingly, the OA-treated tumor spheroids exhibited greater expression of FABP5 and HIF-1α (Fig. 6d) and grew faster compared with vehicle-treated spheroids; this was abrogated by FABP5 or HIF-1α knockdown (Fig. 6e). Immunofluorescence staining for anti-Ki67 further confirmed that OA treatment promoted cell proliferation in spheroids, which was mediated by FABP5 and HIF-1α (Fig. 6f). As HIF-1α induces the expression of genes related with tumor survival, we assessed the mRNA levels of CCND2, VEGF, BNIP3L, and CA9, and found that CCND2, VEGF, BNIP3L, and CA9 were upregulated when spheroids were treated with OA, whereas their mRNA levels decreased upon FABP5 or HIF-1α silencing (Fig. 6g). Collectively, these results indicate that the FABP5/HIF-1α axis is involved in OA-driven HCC cell growth.

**HIF-1α inhibition by vitamin C suppresses OA-induced lipid-droplet formation and cell proliferation**. As vitamin C impacts

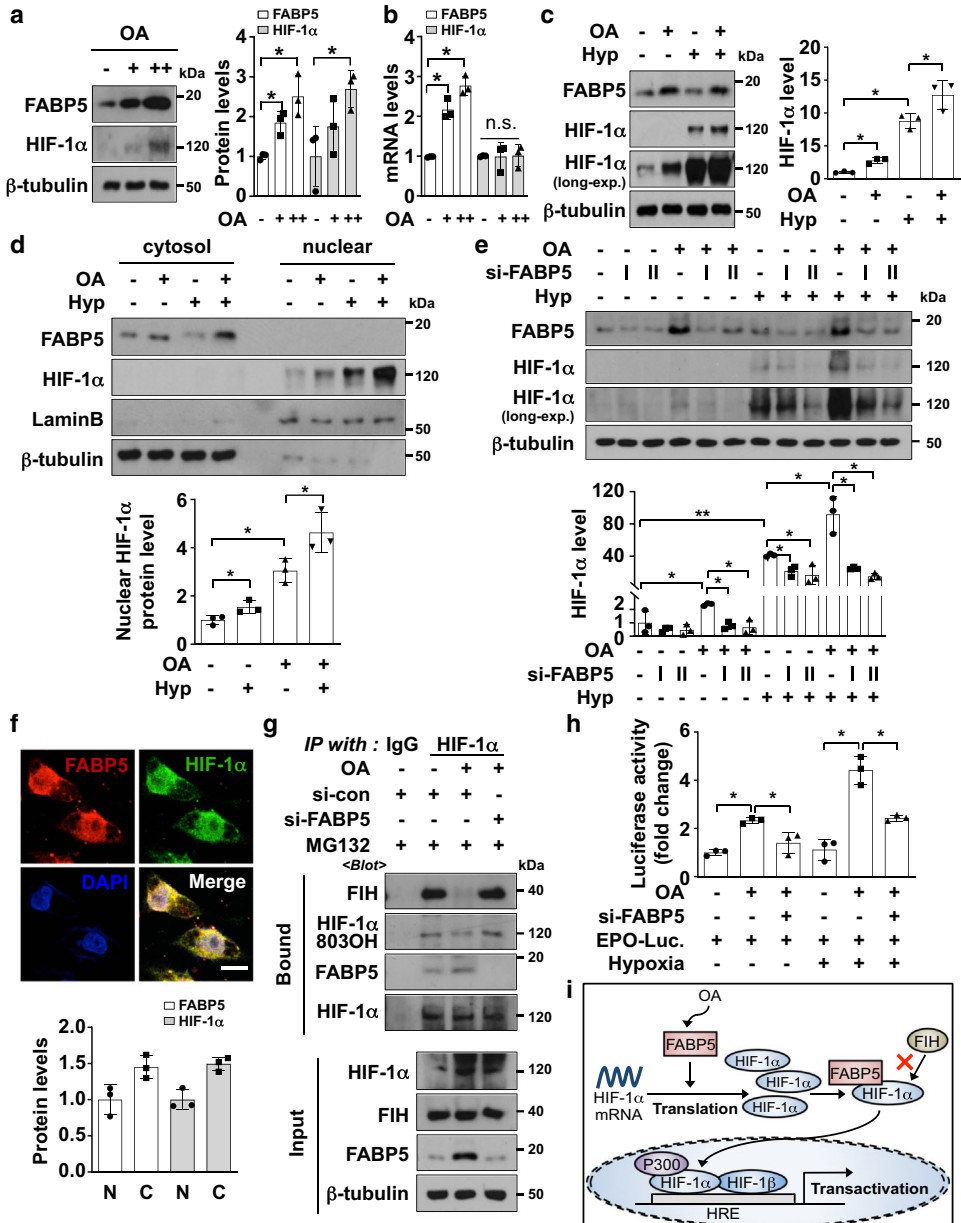

**Fig. 4 Oleic acid (OA)-induced FABP5 upregulation promoted HIF-1α activation in HCC cells. a, b** HepG2 cells were incubated with OA (100 or 200 µM) for 24 h and subjected to western blotting (**a**) or RT-qPCR (**b**). Data are presented as the means ± SD (n = 3). *P < 0.05. **c** HepG2 cells were treated with 200 µM OA for 24 h and then incubated under normoxic or hypoxic conditions for 8 h. Cells were subjected to immunoblotting and the blots were quantified (mean ± SD, n = 3). *P < 0.05. **d** HepG2 cells treated with OA for 24 h were incubated under the indicated conditions for 8 h, and whole-cell lysates were fractionated into cytosolic and nuclear compartments. The fractions were assessed using western blotting and nuclear HIF-1α protein levels were calculated based on Lamin B expression levels (mean ± SD, n = 3). *P < 0.05. **e** HepG2 cells were transfected with si-control or si-FABP5 and then treated with OA for 24 h. Samples were incubated under normoxic or hypoxic conditions for 8 h and then subjected to western blotting. The quantification was performed using Image J (mean ± SD, n = 3). *P < 0.05; **P < 0.0001. **f** Representative immunocytochemistry images. HepG2 cells were incubated with 10 µM MG132 for 8 h and subjected to immunocytochemistry with the indicated antibodies. All samples were also stained with DAPI. N nuclear, C cytosol, Scale bar: 10 µm. **g** HepG2 cells were transfected with si-control or si-FABP5 and incubated with 200 µM OA for 24 h, followed by treatment with MG132 10 µM for 8 h. Lysed proteins were precipitated using IgG or HIF-1α antibody and subjected to immunoblotting. **h** HepG2 cells were co-transfected with EPO-Luc plasmid, CMV-B-galactosidase plasmid, and si-control or si-FABP5, and then treated with 200 µM OA for 24 h. Cells were incubated under the indicated conditions for 16 h and assessed using luciferase assay. Luciferase activities are presented as the means ± SD (n = 3). *P < 0.05 **i** The proposed mechanism by which OA triggers FABP5/HIF-1α pathway activation in cancer cells in the hypoxic tumor microenvironment.

HIF-1α stability, we examined whether vitamin C treatment could inhibit OA-induced lipid-droplet formation and cell proliferation. We found that OA-induced HIF-1α expression could be suppressed by vitamin C treatment under hypoxic conditions (Fig. 7a). We also found that HepG2 cells that were treated with OA and vitamin C formed a decreased number of lipid droplets compared with cells treated with OA only (Fig. 7b), and that the expression of genes involved in lipid accumulation was reduced back to normal levels (Fig. 7c and Supplementary Fig. 5a). Furthermore, OA-induced cell proliferation was suppressed by concomitant treatment with vitamin C; this was confirmed using a colony-formation assay and a 3D tumor spheroid culture (Fig. 7d,

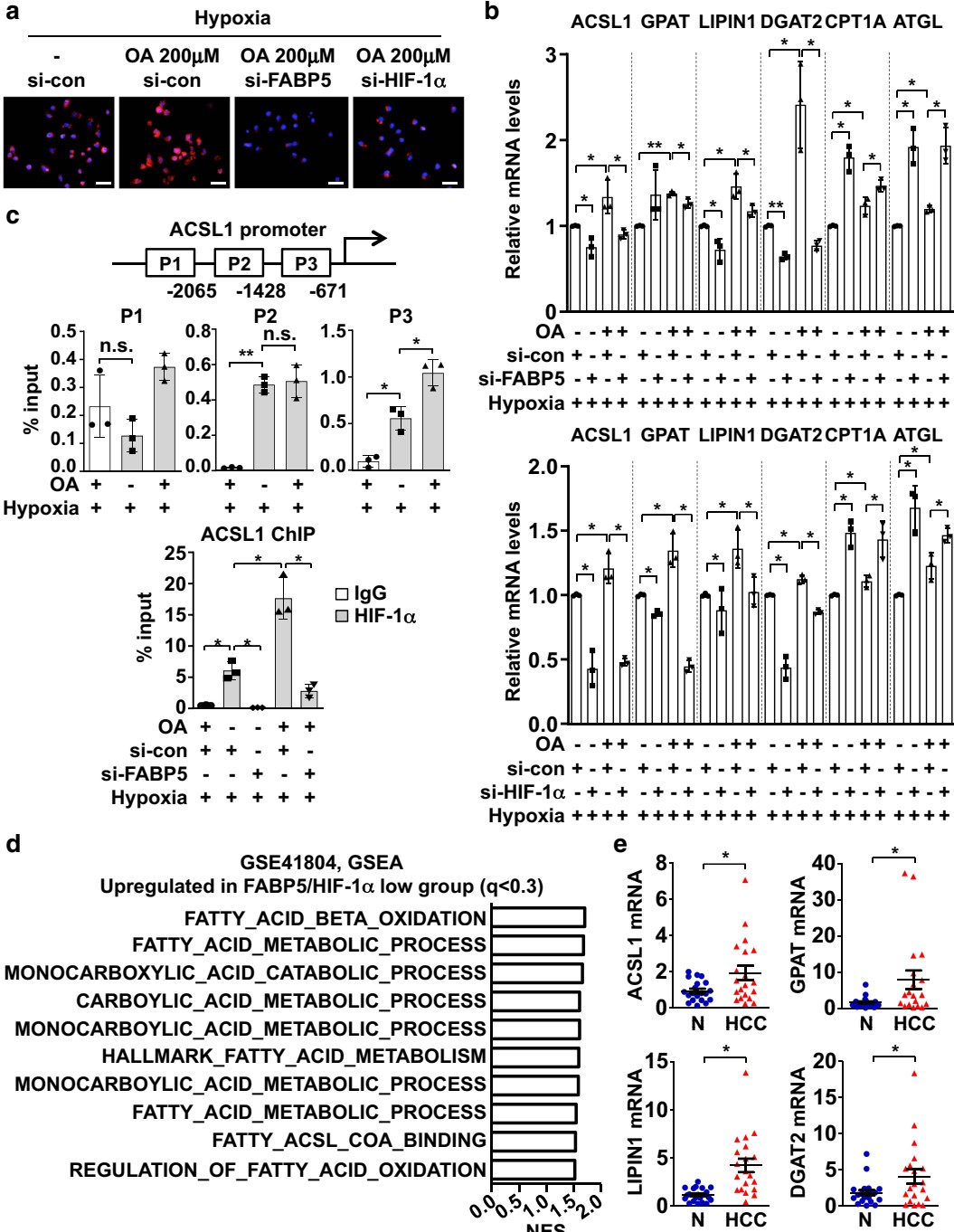

**Fig. 5 The FABP5-HIF-1α axis promotes lipid accumulation in HCC cells. a** Representative Nile-red staining images. HepG2 cells were transfected with si-control or si-FABP5 or si-HIF-1α, and cells were seeded on coverslips and incubated under hypoxic conditions for 8 h. Samples were fixed with 4% paraformaldehyde and stained with Nile red. All samples were also stained with DAPI. Scale bar, 50 μm. **b** The si-FABP5 or si-HIF-1α (or si-control) transfected HepG2 cells were treated with OA for 24 h and incubated under hypoxic conditions for 16 h. Cells were lysed for RNA extraction, and RT-qPCR analysis was used to determine the mRNA levels of genes involved in fatty-acid metabolism. Relative mRNA levels are presented as the means ± SD ($n =$ 3). *$P < 0.05$, **$P < 0.0001$. **c** HIF-1α binding to the ACSL1 promoter region, which contains core DNA sequences for HIF-1 binding (CGTG; P1: −2119, P2: −1491, P3: −730), in HepG2 cells was detected using chromatin immunoprecipitation coupled quantitative PCR using nonimmunized serum (IgG) or anti-HIF-1α. Bars represent the means ± SD ($n =$ 3). *$P < 0.05$; **$P < 0.0001$; n.s. not significant. **d** GSEA results showing that fatty-acid-metabolism-related gene signatures were upregulated in FABP5/HIF-1α-low HCC samples (GSE41804). **e** ACSL1, GPAT, LIPIN1, and DGAT2 mRNA levels in human HCC tissues were analyzed using RT-qPCR. Data are presented as the means ± SEM. *$P < 0.05$.

e). We also found that the upregulation of CCND2, VEGF, BNIP3L, and CA9 after OA treatment was inhibited when spheroids were treated with vitamin C in addition to OA (Fig. 7f).

Taken together, these results suggest that the pharmacological inhibition of HIF-1α by vitamin C can reduce lipid-droplet formation and cell proliferation in HepG2 cells.

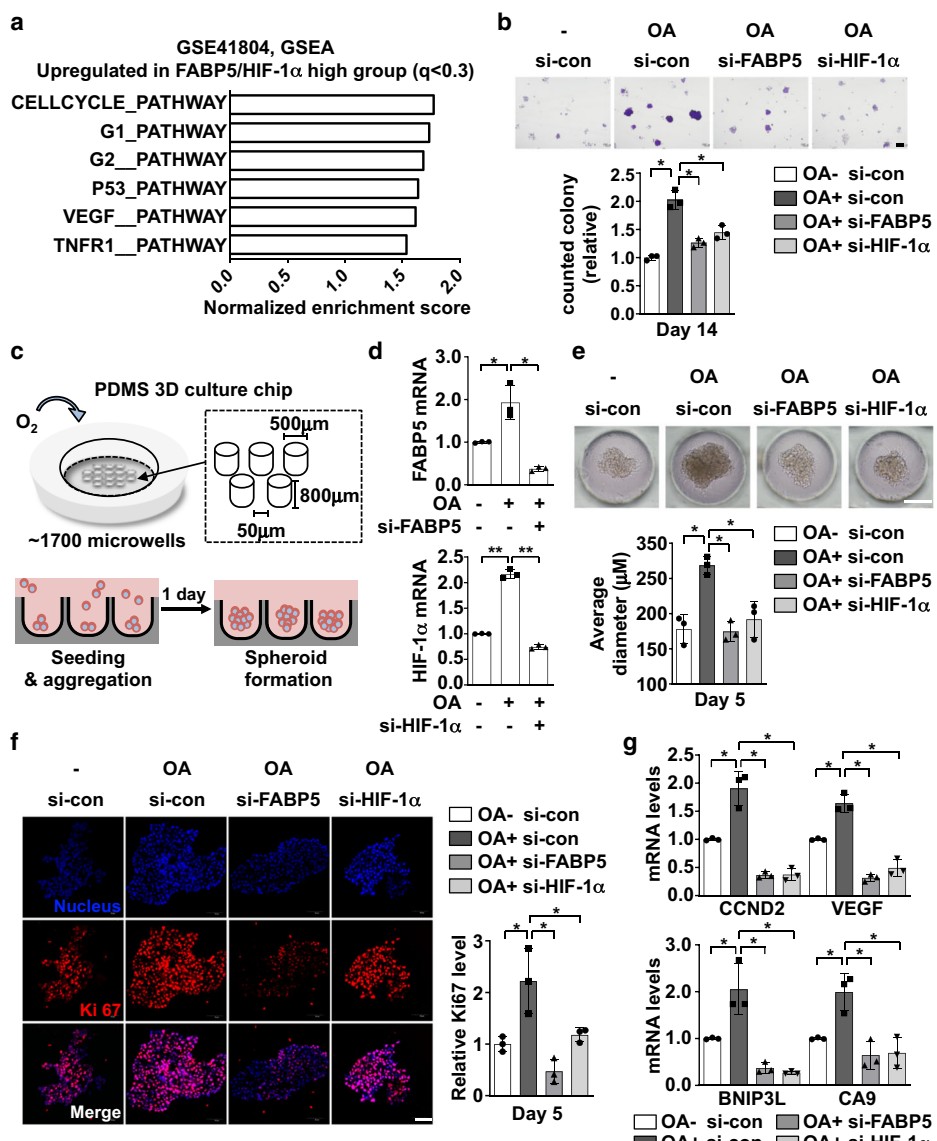

**Fig. 6 The FABP5–HIF-1α axis mediates OA-induced HepG2 cell survival. a** GSEA results showing gene signatures that were upregulated in FABP5/HIF-1α-high HCC samples (GSE41804). **b** HepG2 cells were transfected with si-FABP5 or si-HIF-1α and subjected to colony-formation analysis under OA-high conditions. Colonies were counted and data are presented as the means ± SD. *$P < 0.05$, **$P < 0.0001$. **c** Schematic representation of the polydimethylsiloxane (PDMS) spheroid culture chip and details of the spheroid-culture-chip design. The spheroid-formation process for experiments is indicated (see also Supplementary Fig. 4c). **d** HepG2 cells were transfected with si-FABP5 or si-HIF-1α and subjected to three-dimensional (3D) spheroid culture. RT-qPCR was used to analyze FABP5 or HIF-1α mRNA levels in HepG2 spheroids cultured for 5 days. Relative mRNA levels are presented as the means ± SD ($n = 3$). *$P < 0.05$; **$P < 0.0001$. **e** Representative optical-microscopy images of spheroids on day 5 of culture. The average diameter was calculated using Image J, and data are presented as the means ± SD ($n = 3$). *$P < 0.05$, **$P < 0.0001$ **f** Immunofluorescence staining of HepG2 spheroids cultured in PDMS chips. Staining was performed using 10 μm frozen spheroid sections (red, Ki67; blue, DAPI). Ki67 expression levels were calculated by using Image J (mean ± SD, $n = 3$). Scale bar, 50 μm. *$P < 0.05$. **g** Cultured 3D spheroids were lysed for RNA extraction, and RT-qPCR was used to analyze survival-related genes. Relative mRNA levels are presented as the means ± SD ($n = 3$) *$P < 0.05$.

## Discussion

In response to hypoxic stress, cancer cells express HIF-1α, which plays a pivotal role in cell metabolism, proliferation, and cellular adaptation under hypoxic conditions. In this study, we identified FABP5 as a HIF-1α activator under the influence of OA, and determined that the FABP5/HIF-1α axis regulates lipid metabolism and cell proliferation in HCC. Therefore, we propose that targeting the FABP5/HIF-1α axis can be a promising therapeutic approach for the inhibition of metabolic-reprogramming-driven HCC development and progression.

FABP5 is a well-characterized protein involved in several types of cancer, as it promotes cell proliferation, migration, and

invasion; thus, its expression has been associated with poor cancer prognosis. OA has been proposed as a potential biomarker of HCC, yet its direct binding to FABP5 had not been demonstrated previously. Analysis of the plasma phospholipid fatty-acid composition suggested that HCC patients had higher levels of OA compared with healthy individuals. Additionally, the comparison of fatty-acid distribution in cancerous tissues and their surrounding tissues in HCC patients led to the identification of OA as the most increased fatty acid in HCC patients. These studies suggested that the changes in OA levels observed in HCC patients are the result of intrinsic abnormal fatty-acid metabolism due to cancer pathology and not changes in diet[5,29–34]. Considering the

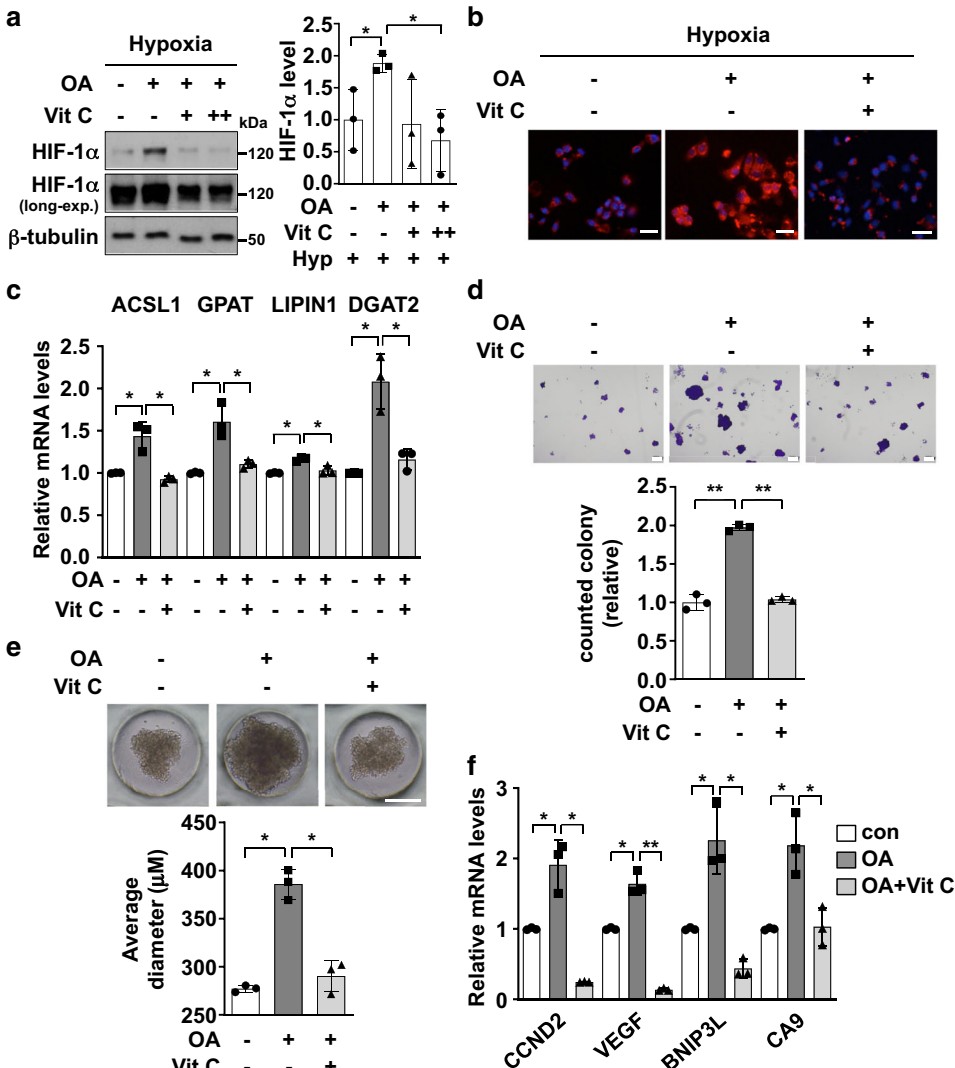

**Fig. 7 HIF-1α inhibition suppresses OA-induced lipid accumulation and cell proliferation. a** HepG2 cells were pre-treated with 200 μM OA or not treated for 24 h. Cells were then treated with 100 μM vitamin C for 24 h and incubated for 8 h under hypoxic conditions. Cell lysates were subsequently subjected to western blotting and HIF-1α expression levels were calculated using Image J (mean ± SD, n = 3). *P < 0.05. **b** Representative images after Nile-red staining in HepG2 cells. OA-treated cells were subjected to Nile-red staining with or without vitamin C under hypoxic conditions. Scale bar, 200 μm. **c** OA-treated and vitamin-C-treated spheroids were lysed for RNA extraction, and RT-qPCR was then used to quantify the mRNA levels of genes related to fatty-acid metabolism. Relative mRNA levels are presented as the means ± SD (n = 3). *P < 0.05 **d** Representative optical images for colony formation by OA-treated HepG2 cells with or without vitamin C. Colony counts are presented as the means ± SD. **P < 0.0001. **e** Representative optical images of HepG2 spheroids treated with OA and vitamin C. The average spheroid diameter was calculated using Image J. Data are presented as the means ± SD. *P < 0.05, **P < 0.0001. **f** Spheroids treated with OA and vitamin C were subjected to RT-qPCR to quantify the mRNA levels of HIF-1α-target genes involved in cell survival. Relative mRNA levels are presented as the means ± SD (n = 3). *P < 0.05, **P < 0.0001.

abnormally high levels of OA in HCC patients, we assessed the impact of OA on the protein profiles in HCC cells and found that high levels of OA upregulated FABP5 expression. Our data indicate that FABP5 is a critical fatty-acid sensor in HCC, as it promotes HCC progression and poor prognosis by activating HIF-1α.

Due to poor vascularization, cancer cells reprogram lipid metabolism to enable cell survival and growth despite the lack of nutrients in the tumor microenvironment[23]. A recent report revealed the contribution of FABP5 to the reprogramming of lipid metabolism in cancer. In aggressive prostate and breast cancer cells, FABP5 knockdown repressed the expression of genes involved in lipolysis, including HSL and MAGL, as well as that of genes involved in de novo FA synthesis, such as ELOVL6 and ACSL1. However, the transcription factor responsible for the

FABP5-mediated changes in the expression of genes involved in lipid metabolism was not identified[1,6–10]. In this study, we demonstrated that FABP5 upregulated the expression of HIF-1α, as well as enhanced its transcriptional activity by promoting the binding of HIF-1α to a HRE in the ACSL1 promoter region. ACSL1 converts fatty acids into fatty acyl-CoA, and subsequently, GPAT, LIPIN1, and DGAT transform fatty acyl-CoA into lipid droplets; thus, ACSL1 plays a crucial role in the initial steps of fatty-acid metabolism in cells. We found that OA-induced FABP5/HIF-1α axis activation promoted the upregulation of ACSL1 as well as genes involved in lipid-droplet formation. HIF-1α is known to regulate fatty-acid synthesis and lipid storage by inducing the expression of LIPIN1 and FASN, respectively[8,35]. Therefore, our work identified ACSL1 as a HIF-1-target gene that drives lipid metabolism. The results also suggest that the FABP5/

HIF-1α axis can be used to modulate lipid metabolism in cancer cells.

Cytosolic fat is stored in the form of lipid droplets, which are involved in energy production, protection against ROS, and membrane biogenesis necessary for rapid cancer cell growth[15,36–41]. Accumulation of lipid droplets has been demonstrated in various types of cancer, including breast, brain, bile-duct, cervical, colon, liver, lung, ovarian, prostate, and pancreatic cancer as well as clear-cell renal cell carcinoma[42–53]. Moreover, several studies reported that there is an association between lipid levels in tumors and poor prognosis. Numerous studies have also suggested that lipids could promote cancer cell proliferation and tumor growth. It has been reported that human colon cancer cell lines contain a remarkably large number of lipid droplets, and that this is associated with enhanced cell proliferation. Similarly, metabolic alterations resulting in impaired lipid droplet formation has been associated with G1 cell cycle arrest. Moreover, OA has been reported to promote the growth of breast cancer cells by activating the PI3K pathway, suggesting that targeting lipid droplet formation in cancer cells can be a useful strategy for treating cancer patients[49,54,55]. In our study, we found that OA could increase HCC cell growth and colony formation in vitro, and these phenomena were repressed by FABP5 or HIF-1α silencing. We also used an in vitro 3D culture system, which has properties that are very similar to those of tumors, including hypoxia-gradients, cell shape, and enhanced chemoresistance. Compared to other conventional 3D spheroid culture systems, the polydimethylsiloxane (PDMS) chip promotes improved cell growth rate and function, as PDMS is oxygen-permeable, allowing for enhanced direct oxygen supply into the spheroids. Furthermore, PDMS microwell structures enable size-regulated aggregation of spheroids, thus offering highly reproducible cell growth[56–58]. Consistent with results from cell counting and colony formation assays, we further confirmed that FABP5 or HIF-1α silencing repressed OA-mediated spheroid growth, suggesting that the OA/FABP5/HIF-1α axis plays a role in HCC cell survival and growth. Finally, we showed that the natural inhibitor of HIF-1α, vitamin C, could repress OA-induced lipid accumulation and cell survival.

In the present study, we firstly revealed two-step hit processes for HIF-1α activation by FABP5; FABP5 enhances (1) translation of HIF-1α (2) transcriptional activation of HIF-1 via inhibiting FIH-mediated HIF-1α hydroxylation and enhancing p300/HIF-1α binding. Interaction between HIF-1α and FIH was interfered with FABP5, which also identified as an interacting partner for N-terminal of HIF-1α. These results led us to hypothesize that FABP5 might change the HIF-1α conformation, thereby HIF-1α/FIH binding was repressed. Additionally, FABP5 might also be related to FIH directly due to FABP5 enhanced Gal4-CAD activity that only contains C-terminal of HIF-1α. However, the effect of FABP5 on HIF-1α full-length activity was higher than those in Gal4-CAD activity, we focused on the role of FABP5 on interrupting HIF-1α/FIH binding. The detail molecular mechanism for the relationship between FABP5 and FIH remains still an open question. In addition, we presented that OA-induced FABP5 reinforced HIF-1α protein and its transcription activity in Fig. 4. As HIF-1 plays in the nucleus, HIF-1α localization is an important checkpoint for a better understanding of HIF-1 activity. Recent studies revealed that nuclear accumulation of HIF-1α is dependent on ERK signaling pathways and HIF-1α binding partners[59,60], localization of HIF-1α might also be affected by OA and FABP5, which is an open question to date.

In conclusion, our results indicate that FABP5-mediated activation of HIF-1α is a HIF-1α activating mechanism induced by OA. The FABP5-mediated HIF-1α activation is achieved through multiple mechanisms, including an increase in protein synthesis,

inhibition of FIH-dependent hydroxylation, and then p300-dependent transcriptional activation. Our findings also provided new insights into the role of OA/FABP5/HIF-1α axis in the lipid-metabolism reprogramming involved in cancer cell growth in HCC. Based on our results, targeting this axis can be a potential strategy for inhibiting metabolic-reprogramming-driven HCC development and progression.

## Methods

**Cell culture.** HepG2 (a human hepatocellular carcinoma cell) cell lines were obtained from the Korea Cell Bank (Seoul, South Korea). HEK293 (a human embryonic kidney cell) cell lines were obtained from the American Type Culture Collection (Manassas, VA, USA). HepG2 and HEK293 cells were cultured in Dulbecco's Modified Eagle's Medium (Welgene, Gyeongsan-si, South Korea) supplemented with 10% fetal bovine serum (Welgene, Gyeongsan-si, South Korea) and 1% penicillin/streptomycin (Thermo, Rockford, IL, USA). Incubator gas tension was maintained at 1% $O_2$/5% $CO_2$ for hypoxic conditions and 21% $O_2$/5% $CO_2$ for normoxic conditions (VS-9000GC; Vision Scientific, Seoul, South Korea).

**Human hepatocellular carcinoma tissues.** For immunohistochemistry, human hepatocellular carcinoma tissue arrays were purchased from SuperBioChips Lab (Seoul, South Korea). For analyzing mRNA levels in human hepatocellular carcinoma tissues, tissues were obtained with consent under approval by the Institutional Review Board (IRB) committees of the Seoul National University Hospital (SNUH). Detailed clinical information for used in this study is summarized in Supplementary Tables 1 and 2.

**Immunohistochemistry in human hepatocellular carcinoma tissues.** The tissue slides were incubated in a 60 °C oven for 1 h to remove paraffin, and were microwaved in antigen retrieval solution for 20 min. After treatment with 3% $H_2O_2$, samples were incubated with primary antibodies (anti-FABP5 or anti-HIF-1α) overnight at 4 °C, followed by biotinylated with a secondary antibody at room temperature for 1 h. The immune complexes were visualized using a Vectastain ABC kit (Vector Laboratories, Burlingame, CA, USA), and tissue slides were counterstained with hematoxylin for 10 min at room temperature. Protein expression levels were evaluated based on intensity and positively stained cell number in four independent high-power fields on each sample.

**Antibodies.** Antibody against FABP5 was purchased from R&D systems (Minneapolis, MN, USA); anti-p-AKT, anti-p-mTOR, and anti-Ki-67 from Cell Signaling (Danvers, MA, USA); anti-Flag from Sigma-Aldrich (St. Louis, MO, USA); anti-B-tubulin and anti-Lamin B from Santa Cruz Biotechnology (Santa Cruz, CA, USA); anti-GFP from Thermo Fisher Scientific (Waltham, MA, USA); anti-HA from GeneTex (Irvine, CA, USA). Anti-HIF-1α was generated against human HIF-1α in rabbits, and a monoclonal anti-hydroxylated Asn [803] of HIF-1α was raised in mouse as described previously[61,62].

**Plasmids and short interfering RNAs (siRNAs).** HA-tagged HIF-1α, GFP-tagged HIF-1α, GFP-tagged fragments of HIF-1α (N-terminal, ODDD and C-terminal), HA-tagged p300, Gal4-CAD (amino acids 776–826 of HIF-1α), Gal4-CAD N803A mutant, VP-16-p300 CH1 plasmids were constructed, as previously described[61,62]. The cDNA of FABP5 was cloned by reverse transcription and PCR, and amplified cDNA was inserted into the Flag/SBP tagged pcDNA3 (Clontech Laboratories). All FABP5 siRNAs and HIF-1α siRNA were synthesized by MBiotech (Gyeonggi-do, South Korea). The sequences for targeting FABP5 (NM_001444) and HIF-1α (NM_001530) are listed in Supplementary Table 3.

**RNA isolation and quantitative RT-PCR.** Total RNA from mouse liver tissues or cultured cells was extracted using TRIzol Reagent (Invitrogen, CA, USA). cDNA synthesis was performed in a EasyScript cDNA Synthesis Kit (Applied Biological Materials Inc., Richmond, Canada). Quantitative real-time PCR on 48-well optical plates was performed with Evagreen qPCR master mix reagent (Applied Biological Materials) in StepOne™ Real-time PCR system (Applied Biosystems, BC, Foster City, CA, USA). The sequences of qPCR primers are summarized in Supplementary Table 4. The mRNA values of targeted genes were normalized to 18S rRNA expression level.

**Western blotting and immunopreciptitation.** Cell lysates in a 2× sodium dodecyl sulfate (SDS) sample buffer were separated on SDS-polyacrylamide gels and transferred to Immobilon-P membranes (Millipore, Billerica, MA, USA). The membranes were blocked with a 5% skim milk dissolved in Tris-saline solution containing 0.1% Tween 20 for 1 h and incubated with a primary antibody (1:1000 dilution) overnight at 4 °C. Membranes were incubated with a horseradish peroxidase-conjugated secondary antibody for 1 h and visualized using an ECL Plus kit (Thermo Fisher Scientific, Waltham, MA, USA).

For analyzing protein interactions, cells were lysed in immunoprecipitation (IP) buffer (5 mM EDTA, 50 mM Tris-Cl, 100 mM NaCl and 1% NP-40) supplemented with protease inhibitor cocktail and phosphatase inhibitors. One milligram of cell lysates were incubated with streptavidin bead or HA bead for 16 h at 4 °C. After bead washing steps, precipitated proteins were eluted by 2× SDS sample buffer and immunoblotted.

**Chromatin immunoprecipitation**. Chromatins were cross-linked with 1% for-maldehyde for 10 min at room temperature and then treated with 150 mM glycine. The fixed HepG2 cells were collected by scraping and centrifuged $1000 \times g$ for 5 min, and pellets were lysed in the FA lysis buffer (50 mM HEPES pH 7.5, 140 mM NaCl, 1 mM EDTA, 1% Triton X-100, 0.1% sodium deoxycholate, 0.1% SDS, and a protease inhibitor cocktail). The lysates were sonicated to chop chromosomal DNAs and spun down by centrifugation. Chromatin complexes were resuspended in a chromatin RIPA buffer (50 mM Tris, pH 8.0, 150 mM NaCl, 2 mM EDTA, 1% NP-40, 0.5% sodium deoxycholate, 0.1% SDS, and a protease inhibitor cocktail). The samples were immunoprecipitated with anit-HIF-1α or control IgG overnight at 4 °C and precipitated with precleaned protein A/G bead for 4 h. After washing beads with low and high salt TE buffers (20 mM Tris, pH 8.0, 0.1% SDS, 1% Triton X-100, 2 mM EDTA, and 150 mM or 500 mM NaCl), the complexes were eluted with an elution buffer (1% SDS, 100 mM NaHCO₃) at 65 °C. DNAs were isolated by phenol–chloroform–isoamyl alcohol (25:24:1) and precipitated using ethanol and glycogen. The extracted DNA was resolved in nuclease-free water and analyzed by RT-qPCR.

**Luciferase assay**. The luciferase reporter genes with hypoxia response element (HRE) of the erythropoietin enhancer or muatated HRE were donated by Dr. Eric Huang (University of Utah). For evaluating the cap-dependent translation activity and IRES-dependent translation activity of HIF-1α, TK-5′-UTR-HIF-1α reporter vector and CMV-GFP-5′-HIF-1α reporter vector were constructed as previously described. Luciferase reporter plasmid and the CMV-B-galactosidase plasmid were co-transfected into the cell. After 48 h of stabilization, luciferase activities were measured using a Lumat LB9507 luminometer (Berthold Technologies, Bad Wildbad, Germany), and the reporter activity was divided by B-galactosidase activity to normalize transfection efficiency.

**Gal4 reporter and mammalian two-hybrid assays**. To measure HIF-1α CAD activity, 293T cells were co-transfected with 100 ng of Gal4-Luc plasmid, 100 ng of Gal4-CAD (or N803A) plasmid, 1 μg of CMV-B-galactosidase plasmid and 1 μg of Flag/SBP-FABP5 plasmid using Lipofectamine 2000 reagent. For mammalian two-hybrid assays, 293T cells were co-transfected with 100 ng of Gal4-Luc plasmid, 100 ng of Gal4-CAD plasmid, 500 ng of CH1-VP16 plasmid, and 1 μg of CBV-B-galactosidase plasmid using Lipofectamine 2000 reagent. After stabilization for 48 h, the cells were incubated under normoxic or hypoxic conditions for 16 h, and luciferase activities in the cell lysates were measured using a Lumat LB 9507 luminometer (Bethold Technologies, Bad Wildbad, Germany). The B-galactosidase activites were determined to normalize efficiency of transfection.

**Cell proliferation assays**. To examine cell proliferation, cell counting and colony formation assay were performed. For cell counting, $1 \times 10^5$ cells were seeded in 6-well plate and incubated under normoxic or hypoxic conditions for indicated days. After incubation, cells were immediately detached and counted with hematocyt-ometer. For colony formation assay, the $5 \times 10^3$ transfected cells were seeded in 6-well and incubated with 2 weeks. After 2 weeks, cells were fixed with 4% for-maldehyde and stained with 0.5% crystal violet in methanol at room temperature for 1 h.

**Lipid droplet staining**. Human hepatocellular carcinoma cells (HepG2) were washed in PBS once and fixed with 4% PFA for 10 min at room temperature. After a wash step with PBS, cells were incubated Nile Red (1 mg/ml) for 20 min at 37 °C and subsequently stained with 4′,6-diamidino-2-phenylinodle (DAPI) for 1 min.

**3D culture and immunofluorescence for sectioned spheroid**. 3D cell culture method is originated from Prof. Fukuda. In brief, $1 \times 10^6$ HepG2 cells were seeded in oxygen permeable PDMS plate coating with 4% fluronic. Cells were incubated in the plate for 5 days and average diameter was analyzed using Image J. Immuno-fluorescence staining was performed for frozen sections of HepG2 3D spheroids. Spheroids were fixed with 4% paraformaldehyde for 30 min at 4 °C, washed three times with PBS, and submerged into 10, 20, and 30% sucrose for 1 h, respectively. After that, the spheroids were embedded in OCT compound (Sakura Finetek, Tokyo, Japan) and stored at −80 °C. Sections were cut at 10 μm thickness, and placed on a glass slide. The sectioned spheroids were washed two times with PBS and incubated with 1% BSA solution for 1 h. After blocking step, primary antibody for Ki-67 was added (1:200 dilution) and kept for overnight. Next, spheroids were washed three times with 0.1% Tween-20 in PBS solution, and added with the secondary antibody (Alexa Flour 568, anti-rabbit, 1:400 dilution) for 1 h at room temperature. After that, spheroids were washed with PBST three times, and the nuclei were stained with DAPI (1:400 dilution) for 5 min at room temperature.

Immunofluorescence images were observed using a confocal microscope (FV3000, OLYMPUS, Tokyo, Japan).

**Informatics analysis**. Hepatic gene expressions in patients with HCC were analyzed with publically available NCBI Gene Expression Omnibus (GEO) dataset (www.ncbi.nlm.nih.gov/geo, GSE41804). FABP5 mRNA expression levels (202345_s_at; corresponding to FABP5) were evaluated between the groups using Mann–Whitney U-test. For GSEA (available from http://www.broadinstitute.org/gsea), the dataset for HCC (GSE41804) was downloaded from Gene Expression Omnibus and analyzed. The median value of the 202345_s_at (corresponding to FABP5) was used as criteria for grouping low FABP5 expression group and high FABP5 expression group. False discovery rate (FDR) q-value less than 0.3 was considered statistically significant.

**Nuclear extraction**. Cells were immediately washed twice by ice-cold PBS (pH 7.8) and scraped off the dishes. By centrifuging (3000 rpm, 5 min, 4 °C), the cell pellet was obtained and subsequently resuspended in 0.3 ml of extraction buffer (20 mM Tris-Cl (pH 7.8), 10 mM KCl, 1.5 mM MgCl2, 0.2 mM EDTA, 0.5 mM DTT, 0.5 mM PMSF, 1 mM Na₃VO₄, 1000× protease inhibitor cocktail). After incubating on ice for 10 min, 0.6% NP-40 was added to the cell pellet and centrifugation (6000 rpm, 5 min, 4 °C) was performed to obtain the cytosolic supernatant. Pellets were resuspended in extraction buffer (5% glycerol, 400 mM NaCl), incubated on ice for 30 min and centrifuged (12,000 rpm, 10 min, 4 °C). After centrifugation, the supernatants were collected as the nuclear fraction. Cytosolic and nuclear proteins were eluted by 4× SDS buffer and subjected to western blotting.

**Statistics and reproducibility**. All data were analyzed using GraphPad Prism 8 software or Microsoft Excel 2011 software and the three independent results were expressed as the means and standard deviations (SD) or standard error (SEM). Two-tailed, paired Student's t-test was performed to compare protein and mRNA expression levels, ChIP-qPCR data, luciferase activities, counted colony and cell numbers, and average diameter for spheroids. Mann–Whitney U-test was used to compare protein and mRNA levels of human HCC. Differences were considered statistically significant when P-values were less than 0.05.

**Reporting summary**. Further information on research design is available in the Nature Research Reporting Summary linked to this article.

## Data availability
All data generated or analyzed during this study are included in this published article (and its supplementary information files). All source data underlying the graphs in this study are available at Supplementary Data 1. Raw data file for LC-MS/MS is included in Supplementary Data 2. Full blots are shown in Supplementary Information.

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

## Acknowledgements

The authors thank Dr. Eric Huang (University of Utah) for kindly giving HRE-Luc, mutated HRE-Luc, Gal4-Luc, and Gal4-CAD plasmids. This work was supported

by grants from the National Research Foundation of Korea (2016R1A2B4013377, 2018R1A2B6007241, 2019R1A2C2083886, 2018R1A5A2025964). J.S. and D.W.J. received a scholarship from the BK21-plus program, Republic of Korea.

## Author contributions

J.S. performed most of the experiments and participated in conception and design, analyzed data and drafted the manuscript with Y.S.C., D.W.J., G.W.L., J.F., and J.W.P. offered materials and participated in experimental design. Y.S.C. conceived and designed experiments, edited the manuscript, and supervised J.S.

## Competing interests

The authors declare no competing interests.
