## [Peer Review File · Communications Biology]

Reviewers' comments:

Reviewer #1 (Remarks to the Author):

The manuscript by Seo et al. identifies FABP5 as a HIF-1a binding partner and demonstrates that oleic acid and FABP5 enhance HIF-1a expression to drive cancer cell proliferation under normoxic and hypoxic conditions. Overall, the manuscript is novel and the data support the overall conclusions. However, I have several concerns that should be addressed prior to publication.

- 1) Fig 1A describes the results of the pulldown experiment. However, the origin of the proteins (or lysates?) used for this experiment is not stated in the text nor in the figure legend. Are these cell lysates?
- 2) Fig 2A shows data from western blots and the authors conclude that that FABP5 increases HIF-1a expression. However, quantification of these blots and statistical analyses of HIF-1a expression are not presented. Thus, it is unclear whether a statistically significant increase is in fact observed. A similar lack of quantification and statistical analysis is present in other figures in the manuscript and must be added.
- 3) Fig 2E: Given that hypoxia increases HIF-1a expression, it is unclear why the HIF1A luciferase activity is lower under hypoxic conditions compared to normoxia. Please clarify.
- 4) Fig 3A and Page 6 line 148: Please state in the text that FABP5 failed to increase activity of the mutated promoter. Also, describe the mutated promoter (i.e., what is mutated).
- 5) The data in Fig 3C are confusing and the individual constructs that are used are not clearly defined. The figure legend is also lacking such key information.
- 6) The blot in Fig 4E is of poor quality and FABP5 knockdown is not quantified. There is also a discrepancy wherein siII appears to possess greater efficacy in suppressing HIF-1a expression compared to siI. However, siI appears to knockdown FABP5 more efficiently. This discrepancy should be explained, quantification of the blots presented, and ideally a more convincing blot presented.
- 7) Fig 6F: The Ki67 images should be quantified and statistically analyzed.
- 8) It is unclear why Fig 7 is included in the manuscript as the relationship between vitamin C and FABP5 is not explored.

Reviewer #2 (Remarks to the Author):

Seo et al 2020 manuscript

Date: 26/04/2020

The manuscript by Seo and colleagues describes how in hepatocellular carcinomas fatty acid binding protein 5 (FABP5) interacts with and enhances hypoxia inducible factor 1 (HIF1) activity, and that this activity is directly influenced by the monounsaturated fatty acid oleic acid. The manuscript is well written and the experiments robust, but currently there are four primary issues with the paper: 1) Conceptually, how are oleic acid levels rising if the cells are hypoxic? Would we not expect less penetrance of lipids into the tumour microenvironment due to reduced vasculature? 2) All experiments have been performed using overexpression systems and endogenous protein interactions have not been shown. 3) There is no data to show that the histological tools used to detect HIF1a and FABP5 protein expression have been optimised. 4) There is no in vivo tumour kinetics data to show the authors model is correct. I have extended some of these points below.

The authors cite the literature that oleic acid levels are higher in HCC than normal tissue but conceptually if tumours are hypoxic then the contribution of lipids from the diet will be reduced due to deficient vasculature. This means that the oleic acid must be being produced by de novo lipogenesis, but you have to be aware that lipid desaturation is an oxygen dependent process so

its levels might go down in hypoxia. The authors need to show in their cell models that putting the cells under hypoxia increases oleic acid levels and decreases palmitate/stearate levels (Peck & Schulze 2016 Lipid desaturation – the next step in targeting lipogenesis in cancer?). Indeed, calculating the ratio of these lipids is likely key to how HIF1a is activated. Taking an example from TCA metabolism, it is the ratio of aKG/succinate ratio that impacts transcriptional activity not the absolute levels (Morris JP et al 2019 α -Ketoglutarate links p53 to cell fate during tumour suppression).

There is no evidence that this interaction occurs with endogenous protein and that FABP5 competes with other mechanism to regulate HIF1 protein activity. The authors should put the cells under hypoxia and show FABP5 binds to endogenous HIF1. Moreover, silencing of FABP5 and a clear reduction of HIF transcriptional activity is required.

There is no data to show that these antibodies have been previously optimised i.e. on cell lines treated with siRNA or CRISPR knockout? In the normal tissue HIF1a expression looks to be at the extracellular membrane, can the authors explain this as HIF1 is normally in the nucleus? Also, the FABP5 expression in the tumour tissue looks diffuse yet the authors posit that these proteins are frequently interacting. Shouldn't at least a proportion of the protein subcellular localisation be similar if they are directly interacting?

The authors need to show that they have efficacy in vivo with models that are depleted in FABP5 expression (shRNA/CRISPR). If a difference in size is observed then the authors can perform RNAseq or RT-qPCR to show that the expression of HIF1a downstream targets are reduced, validating their hypothesis

Minor suggestions:

- There should be a supplemental table including all the proteins that were identified to interact with HA-HIF1.
- There is no explanation what F/S-FABP5 means. I presume full length FLAG FABP5?
- Reference 25 should be added to the introduction to make it clear that it is known that oleic acid levels are increased in HCC.
- Line 44 - "malignant cancer cells increase exogenous fatty acids uptake". In general, malignant cells increase de novo lipogenesis. Hence the increase in FASN in multiple cancers. This should be represented in the text.
- Line 283-4 – these references should be used in the introduction to introduce lipid synthesis in HCC better.
- Line 305 – cite the Jain et al 2020 (Genetic Screen for Cell Fitness in High or Low Oxygen Highlights Mitochondrial and Lipid Metabolism) that dependency of cells on lipid synthesis increases under hypoxia.
- The authors could immunoprecipitate FABP5 and show that this is a HIF1a specific phenomenon i.e. it does not interact with HIF2a

Reviewer #3 (Remarks to the Author):

In the present manuscript entitled "Fatty-acid-induced FABP5/HIF-1 reprograms lipid metabolism and enhances cell proliferation" the authors identify FABP5 as a HIF-1 α interacting protein. Using cell transfection and overexpressed proteins, they find that FABP5 interacts with the N-terminus of HIF-1 α , and enhances HIF-1 α synthesis and transcriptional activity. They propose that the latter effect is mediated by inhibition of HIF-1 α hydroxylation by FIH and enhanced binding of p300. They further show that both FABP5 and HIF-1 α are highly expressed in hepatocellular carcinoma tissues and cell lines and that treatment of cells with oleic acid upregulates FABP5 and HIF-1 α ,

induces the expression of genes involved in lipid metabolism and the hypoxic response. Moreover, they show that under these conditions FABP5 and HIF-1 α promote lipid accumulation and cell proliferation in HCC cells.

Recent studies have identified a growing number of genes involved in different aspects of lipid metabolism as HIF-1 targets affecting cancer cell proliferation, survival and chemoresistance. The novel findings presented here that FABP5 is also part of this network and that HIF-1 α expression and activity are induced by fatty acids are overall very interesting. However, all the experiments showing a physical interaction between HIF-1 α and FABP5 were performed with overexpressed proteins. The authors must confirm that this interaction also occurs between endogenous proteins. In addition, the findings of some of the experiments describing the effect of FABP5 on HIF-1 α need to be confirmed by the addition of controls.

Specific comments

1. All the experiments showing a physical interaction between HIF-1 α and FABP5 were performed with overexpressed proteins. The interaction between endogenous proteins must be confirmed.
2. Figure 1a. The authors have used as bait only the N-terminal domain of HIF-1 α . They should explain the rationale of this choice and give more information on the construct they have used.
3. Fig. 1b. Transfection with an empty FLAG vector should be included as negative control.
4. ARNT also binds to the N-terminal region of HIF-1 alpha. The authors should check if ARNT and FABP5 compete for binding to HIF-1 α .
5. In Fig. 2a overexpressed FLAG-FABP5 is found both in the nucleus and the cytoplasm. In Fig. 4d endogenous FABP5 is only detected in the cytoplasm. Mislocalization could result from overexpression. Experiments with endogenous proteins can help clarify this issue. In addition, proximity ligation assays could be also informative.
6. Is FABP5 expression regulated under hypoxia and/or by HIF-1?
7. The authors should test if the effect of FABP5 on tumor growth is more severe under hypoxia than under normoxia.
8. Fig. 3f and 4g. If FABP5 interacts with the N terminal HIF-1 α domain, how do the authors explain its effect on Gal4-CAD which only contains the HIF-1 α CAD? It is possible that FABP5 interacts directly with FIH. The authors should test this in their experiments.
9. Fig. 3c. A more detailed description of the constructs is needed. It is not clear from the figure or the text, which parts of HIF-1 α aa sequence are present in each one of them. The MW of the GFP inputs should be indicated on the western blot. Empty FLAG vector controls should be included.
10. Fig. 3e. The authors should analyze the presence of FABP5 in the immunoprecipitates. In addition, they should show the effect of FABP5 overexpression and FABP5 silencing on the interaction between FIH and endogenous HIF-1 α .
11. Fig. 4e and 4f. The increase of HIF-1 α levels shown at Fig. 4e are not in agreement, as should be expected, with a corresponding increase in HIF-1 activity in Fig. 4f. The authors should comment on this discrepancy.
12. Fig. 5c. The sequence of the functional HRE should be shown. The sequence of the primers used for the CHIP should be added to the supplementary figures.
13. The authors should include the following information:
 - The authors should add in the introduction more information regarding the effect of HIFs on lipid metabolism and the effect on cancer cell. A relative review was published recently (Cells 2019, 8, 214; doi:10.3390/cells8030214).
 - FABP3, 4 and 7 have been previously shown to be induced by HIF-1 and should be discussed and cited (Cell Rep. 2014, 9, 349–365 and J. Hepatol. 2015, 63, 855–862).
 - LIPIN1, CPT1A and ATGL have been previously shown to be regulated by hypoxia and HIFs. The respective papers should be cited (Mylonis et al, J. Cell Sci. 2012, 125, 3485–3493; Liu et al, Toxicol. Lett. 2014, 226, 117–123; Du et al, Nat. Commun. 2017, 8, 1769; Zhang, X. et al. eLife 2017, 6)
14. Error bars appear to be missing or abnormally small in Figures 3b, 3d (right panel), 4b, 5b, 6c, 6g, 7c and 7f.

Your manuscript entitled "Fatty-acid-induced FABP5/HIF-1 reprograms lipid metabolism and enhances cell proliferation" has now been seen by 3 referees, whose comments are appended below. You will see from their comments copied below that while they find your work of considerable potential interest, they have raised quite substantial concerns that must be addressed. In light of these comments, we cannot accept the manuscript for publication, but would be interested in considering a revised version that addresses these serious concerns.

We therefore invite you to revise and resubmit your manuscript, taking into account the points raised. Please highlight all changes in the manuscript text file.

To improve this paper, additional experiments and explanations are required and we suggest that you focus on addressing the questions below in your revised version:

(1) Clarify why the HIF1A luciferase activity is lower under hypoxic conditions compared to normoxia. (reviewer 1-FABP5 and cancer)

➔ We appreciate the editorial comments and we also think this point is very important. We measured translation rate of HIF1A by two reporter systems; 5'-cap dependent and IRES-dependent translation (Lee, Shin, Shin, Chun and Park, 2018). Physiologically, the canonical protein synthesis (5'-cap dependent translation) pathway is attenuated due to limited energy availability under hypoxia; we think this also affects 5'-cap dependent HIF1A translation activity (Fig. 2e). Meanwhile, another translational initiation machinery, which is known as internal ribosome entry sites (IRES)-dependent translation, could be an alternative for main translation machinery under hypoxia since this machinery is activated in response to stress (Fig. 2g, Sriram, Bohlen and Teleman, 2018). Therefore, lower 5'-cap dependent HIF1A translation under hypoxic condition is consistent with previous literatures. We hope the reviewer would be satisfied with our answers.

(2) Explain the discrepancy between quantification of the blots presented, and it would be better to add a more convincing blot. (reviewer 1- FABP5 and cancer)

➔ We performed new experiments on Fig. 4e to add more convincing blot. Moreover, we calculated the nuclear HIF-1 α protein level based on Lamin B

expression, and analyzed data was plotted on Fig. 2a with mean value and error bars (n = 3). Furthermore, other blots (Fig. 2f, 4a, 4c, 4d, 4e, and 7a) were also quantified and the significance was calculated; *: $p < 0.05$ and **: $p < 0.0001$.

(3) *Quantify and statistically analyze the Ki67 images. (reviewer 1- FABP5 and cancer)*

➔ We used Image J to quantify the Ki67 images using the formula; Integrated density – (Area of selected area x Mean fluorescence of background readings), <https://theolb.readthedocs.io/en/latest/imaging/measuring-cell-fluorescence-using-imagej.html>. We presented the results next to the representative images in Fig. 6f.

(4) *Explain how are oleic acid levels rising if the cells are hypoxic? (reviewer 2- lipid metabolism and cancer).*

➔ This point is very important and we thank the reviewer for the valuable comments, which have significantly improved the quality of our manuscript. Liver is the major organ to get supplementation of fatty acids to produce glucose through gluconeogenesis under stress condition. For rapid cancer progression, cancer cells require tremendous fatty acids, and it could be fulfilled by de novo synthesis and uptake of exogenous fatty acids. Among de novo synthesis steps, a key enzyme, Fatty acid synthase (FAS) plays important role, which transforms Malonyl CoA to saturated fatty acids. In general, FASN-mediated de novo lipogenesis is accelerated in various types of cancers and FASN expression levels are up regulated by SREBP-1 and HIF-1, thereby promoting saturated fatty acid synthesis (Furuta et al., 2008). Next, saturated fatty acids are folded into unsaturated fatty acids by stearoyl-CoA desaturase 1 (SCD1). However, SCD1 activity requires oxygen, which means unsaturated fatty acids could not be produced by de novo synthesis under hypoxia. Therefore, hypoxic cancer cells depend on exogenous fatty acid uptake to fulfill unsaturated lipid deficiency. As the reviewer mentioned, abnormal blood vessels restrain the delivery of lipids from nutrients. One possibility is that fatty acids are supplied from tumor microenvironment, which is neighboring to cancer cells such as adipose tissue. Pathogenesis and progression of Hepatocellular carcinoma (HCC) is associated with the alterations in fatty acid profile of liver fat. In HCC, cancerous tissue has

significantly higher oleic acid level than its surrounding non-cancerous tissue from same patient, which indicate changes in fatty acid is mainly occurred in tumor microenvironment, but not malnutrition (Qiu, Zhang, Zhang, Hu, Li, Shang and Wan, 2015). Next, exogenous fatty acids are imported into cancer cells by the channel for cellular fatty acid uptake (CD36), which expression is enhanced several cancer types including breast, liposarcoma, and prostate cancer (Kuemmerle et al., 2011). Moreover, hypoxia increases CD 36 expression and function (Mwaikambo, Yang, Chemtob and Hardy, 2009). In conclusion, we think hypoxia enhances the demand for exogenous fatty acid uptake and tumor microenvironment acts as a supporter for providing oleic acids to cancer cells, thereby promoting cancer cell growth. Related recent research have revealed this concept, which tumor lipid metabolism is regulated not only by tumor itself but also by the availability of lipid in tumor microenvironment (Nieman et al., 2011).

(5) Show the data that the histological tools used to detect HIF1 α and FABP5 protein expression have been optimised. (reviewer 2- lipid metabolism and cancer)

➔ For detecting HIF-1 α , the same anti-HIF-1 α antibody as previously described (Chun, Choi, Kim, Choi, Kim, Lee, Kim and Park, 2000) was used. We are sorry for our carelessness that we missed the information related to HIF-1 α antibody; we wrote it to the references (Reference 59, 60). For optimising HIF-1 α expression, si-HIF-1 α transfected HepG2 cells were incubated under hypoxia and immunoblotted with anti-HIF-1 α antibody. The results revealed that specificity of HIF-1 α antibody for detecting HIF-1 α protein (Supplementary Fig. 3b). For detecting FABP5, antibody against FABP5 (R&D systems) was used and confirmed with FABP5 knockdown cells. The results indicated that FABP5 antibody could successfully detect FABP5 protein, which showed decrease according to si-FABP5 treatment (Fig. 4e).

(6) It will be better to show in vivo tumour kinetics data to further approve the model. (reviewer 2- lipid metabolism and cancer)

➔ We thank the reviewer for the thoughtful comments. In fact, a supply of mice and resources related to in vivo experiments are limited these days due to unfortunate circumstances regarding COVID-19. We agreed to the reviewer's comments and that

was the reason for us to utilize *in vitro* 3D culture systems. Especially, PDMS based 3D culture chip has more *in vivo* tumor-like properties including hypoxia-gradients, cell shape and growth, and enhanced chemoresistance compared to conventional 3D spheroid cultures. With PDMS culture chip, we found that OA-mediated spheroid growth was repressed by FABP5 or HIF-1 α silencing, which was consistent with results from 2-dimensional assays (cell counting and colony formation assays). We admit the limitations of 3D cultures; lacking *in vivo* kinetics; however, we perform the utmost methods to approve OA/FABP5/HIF-1 α under this global unfortunate circumstances. We will perform *in vivo* experiment in next project and hope the reviewers understand this situation and would be satisfied with our answers.

(7) Explain the rationale of this choice and give more information on the construct they have used in Figure 1a. Since the authors have used as bait only the N-terminal domain of HIF-1 α . (reviewer 3- FABP5/HIF-1)

➔ The N-terminal domain of HIF-1 α (amino acids 1-400), which we have used as bait to identify HIF-1 α -interacting proteins, contains basic helix-loop-helix (bHLH) domain and PER-ARNT-SIM (PAS) domain. bHLH and PAS domains are a conserved protein domain structure in HIF family, which are essentially required for DNA binding and heterodimerization. Other domains including oxygen-dependent degradation (ODD) domain and C-terminal domains are well known to control the stability and activity of HIF-1 α under normoxic conditions; these domains bind with PHD for ODD domain and FIH for C-terminal domain respectively, and then undergo hydroxylation. The importance of N-terminal domain of HIF-1 α has revealed by literatures, however, little is discovered for N-terminal domain-interacting proteins. Therefore, we sought to focus on HIF-1 α N-terminus and we found FABP5 as HIF-1 α interacting protein. We hope the review would be satisfied with our explanation.

(8) Do experiments with endogenous proteins to clarify the issue that in Fig. 2a overexpressed FLAG-FABP5 is found both in the nucleus and the cytoplasm and in Fig. 4d endogenous FABP5 is only detected in the cytoplasm. Mislocalization could result from overexpression. In addition, proximity ligation assays could be done. (reviewer 3- FABP5/HIF-1)

➔ This is a very important and critical point. First, we confirmed Fig. 2a and Fig. 4d with three independent experiments. As the reviewer mentioned, overexpressed Flag/SBP-FABP5 is found both in the nucleus and the cytoplasm, while endogenous FABP5 is detected only in the cytoplasm. We think these results are due to limitations of nuclear extraction and western blotting, this prompt us to perform immunocytochemistry, which is more sensitive to detect localization of protein. The results revealed that endogenous FABP5 is found both in the cytoplasm and nucleus (but mostly in cytoplasm), and HIF-1 α co-exists with FABP5 in the cytoplasm and nucleus as indicated yellow color (Fig. 4f). Moreover, we found endogenous HIF-1 α binds to FABP5 and this effect reinforced with OA treated HepG2 cells (Fig. 4g). HIF-1 α hydroxylation at Asn803 by FIH is well known as transcriptional inactivation. Although we couldn't afford to conduct proximity ligation assays, we firmly believed that FABP5 binding to HIF-1 α in the cytoplasm prevents binding between HIF-1 α and FIH, thereby HIF-1 α hydroxylation by FIH was reduced (Fig. 3e, 3f as new data). Therefore, oleic acid-induced HIF-1 α becomes more active under normoxia and hypoxia (Fig. 3g, 3h, Fig. 4g and 4h).

(9) Analyze the presence of FABP5 in the immunoprecipitates in Fig. 3e. In addition, they should show the effect of FABP5 overexpression and FABP5 silencing on the interaction between FIH and endogenous HIF-1 α . (reviewer 3- FABP5/HIF-1)

➔ Thanks for a thoughtful suggestion. We added the blot, which indicates the FABP5 in the immunoprecipitates; IP: HA, IB: Flag (FABP5). In addition, we performed the new experiments that reveal the effect of FABP5 overexpression and FABP5 silencing on the binding between endogenous FIH and HIF-1 α in the Fig. 3f and 4g, respectively. As expected, ectopic overexpression of FABP5 attenuates the interaction between FIH and HIF-1 α . Moreover, OA-induced FABP5 also decreases the interaction between FIH and HIF-1 α ; however, si-FABP5 treatment reverses this phenomenon.

Point-by-point responses to the referees' comments:

Reviewer #1 specific comments:

1) *Fig 1A describes the results of the pulldown experiment. However, the origin of the proteins (or lysates?) used for this experiment is not stated in the text nor in the figure legend. Are these cell lysates?*

→ Thanks for your valuable comments. We stated the details of Fig.1a in figure legends and methods. Please see manuscript page 28: line 15-17 and page 18: line 8-12.

2) *Fig 2A shows data from western blots and the authors conclude that that FABP5 increases HIF-1a expression. However, quantification of these blots and statistical analyses of HIF-1a expression are not presented. Thus, it is unclear whether a statistically significant increase is in fact observed. A similar lack of quantification and statistical analysis is present in other figures in the manuscript and must be added.*

→ We quantified the western blots in main and supplementary figures and statistical analysis was performed using GraphPad Prism 8 software and Microsoft Excel 2011. All source data underlying the graph in the study are available at Supplementary information.

3) *Fig 2E: Given that hypoxia increases HIF-1a expression, it is unclear why the HIF1A luciferase activity is lower under hypoxic conditions compared to normoxia. Please clarify. (same as Rebuttal letter Page 1, Q1)*

→ We appreciate the reviewer's comment and we also think this point is very important. We measured translation rate of HIF1A by two reporter systems; 5'-cap dependent and IRES-dependent translation (Lee, Shin, Shin, Chun and Park, 2018). Physiologically, the canonical protein synthesis (5'-cap dependent translation) pathway is attenuated due to limited energy availability under hypoxia; we think this also affects 5'-cap dependent HIF1A translation activity (Fig. 2e). Meanwhile, another translation initiation machinery, which is known as internal ribosome entry sites (IRES)-dependent translation, could be an alternative for main translation

machinery under hypoxia since this machinery is activated in response to stress (Fig. 2g, Sriram, Bohlen and Teleman, 2018). Therefore, lower 5'-cap dependent HIF1A translation under hypoxic condition is consistent with previous literatures. We hope the reviewer would be satisfied with our answers.

4) *Fig 3A and Page 6 line 148: Please state in the text that FABP5 failed to increase activity of the mutated promoter. Also, describe the mutated promoter (i.e., what is mutated).*

➔ Thank you for your valuable comment. We stated these in the manuscript Page 7 line 2-4.

5) *The data in Fig 3C are confusing and the individual constructs that are used are not clearly defined. The figure legend is also lacking such key information.*

➔ 293T cells were transfected with the F/S-FABP5 along with GFP-HIF-1 α -N-terminal (containing aa: 1-400) or GFP HIF-1 α -ODDD (aa: 300-600) or GFP-HIF-1 α -C-terminal (aa: 600-800). We inserted description of individual constructs into figure legends.

6) *The blot in Fig 4E is of poor quality and FABP5 knockdown is not quantified. There is also a discrepancy wherein siII appears to possess greater efficacy in suppressing HIF-1 α expression compared to siI. However, siI appears to knockdown FABP5 more efficiently. This discrepancy should be explained, quantification of the blots presented, and ideally a more convincing blot presented.*

➔ We performed new experiments on Fig. 4E to add more convincing blot. Moreover, the discrepancy was analyzed with three-individual experiments using Image J.

7) *Fig 6F: The Ki67 images should be quantified and statistically analyzed.*

→ Thanks for your suggestion. We quantified the Ki67 images and data was analyzed using Student's t-test.

8) *It is unclear why Fig 7 is included in the manuscript as the relationship between vitamin C and FABP5 is not explored.*

→ Through Fig. 1 to 6, we firmly have suggested that OA/FABP5/HIF-1 α axis is relevant axis for enhancing HCC cell proliferation. Therefore, we think that blocking the axis by chemicals could be effective treatment for HCC and vitamin C is utilized in the study. As reviewer mentioned, the relationship between vitamin C and FABP5 is not explored because we focused on the impact of vitamin C on HIF-1 α stability. Vitamin C is a cofactor of HIF-1 α hydroxylation by PHD, which is preceded ubiquitination by VHL. Therefore, vitamin C is commonly used to induce HIF-1 α degradation. Vitamin C treatment attenuated OA-dependent HIF-1 α induction (Fig. 7a). We also know that OA-dependent ACSL1, GPAT, LPIN1 and DGAT2 induction is HIF-1 α -dependent (Fig. 7c). Therefore, vitamin C treatment is helpful to understand molecular mechanism of OA-dependent HIF-1 α induction and OA-induced transcriptional activity of HIF-1. We hope the reviewer satisfied with our answers.

Reviewer #2 specific comments:

• *There should be a supplemental table including all the proteins that were identified to interact with HA-HIF1.*

→ Thanks for your suggestion. We added all the proteins that were identified to interact with HA-HIF-1 α in Supplemental data Fig. 1a.

• *There is no explanation what F/S-FABP5 means. I presume full length FLAG FABP5?*

→ We are sorry for our carelessness that we missed the full name of F/S-FABP5

(Flag/Streptavidin-Binding Peptide-FABP5, Flag/SBP-FABP5); we wrote it page 4, line 23.

- *Reference 25 should be added to the introduction to make it clear that it is known that oleic acid levels are increased in HCC.*

➔ We added reference 25 in introduction (due to reference numbering system, reference 25 has changed into reference 5).

- *Line 44 - “malignant cancer cells increase exogenous fatty acids uptake”. In general, malignant cells increase de novo lipogenesis. Hence the increase in FASN in multiple cancers. This should be represented in the text.*

➔ Thanks for your thoughtful suggestion. We stated malignant cancer cells increase de novo lipogenesis in introduction (page 2, line 20).

- *Line 283-4 – these references should be used in the introduction to introduce lipid synthesis in HCC better; Line 305 – cite the Jain et al 2020 (Genetic Screen for Cell Fitness in High or Low Oxygen Highlights Mitochondrial and Lipid Metabolism) that dependency of cells on lipid synthesis increases under hypoxia.*

➔ We added the references as the reviewer suggested reference 8 and 23.

- *The authors could immunoprecipitated FABP5 and show that this is a HIF1a specific phenomenon i.e. it does not interact with HIF2a*

➔ Thanks for your comment. However, we focused on the relationship of HIF-1 α /FABP5 since our research had started with HIF-1 α -interactome analysis at the first time. We hope the reviewer understands our initial purpose and accepts our explanation. We'll follow reviewer's comment in the next project.

Reviewer #3 specific comments:

1. *All the experiments showing a physical interaction between HIF-1 α and FABP5 were performed with overexpressed proteins. The interaction between endogenous proteins must be confirmed.*

➔ We appreciate the reviewer's comment and we also think endogenous immunoprecipitation is important. We analyzed the interaction between endogenous HIF-1 α and FABP5 in Fig. 4g. The result revealed that endogenous interaction of HIF-1 α and FABP5 was enhanced by OA treatment and the interaction was attenuated via FABP5 knockdown.

2. *Figure 1a. The authors have used as bait only the N-terminal domain of HIF-1 α . They should explain the rationale of this choice and give more information on the construct they have used. (same as Rebuttal letter Page 4, Q7)*

➔ The N-terminal domain of HIF-1 α (amino acids 1-400), which we have used as bait to identify HIF-1 α -interacting proteins, contains basic helix-loop-helix (bHLH) domain and PER-ARNT-SIM (PAS) domain. bHLH and PAS domains are a conserved protein domain structure in HIF family, which are essentially required for DNA binding and heterodimerization. Other domains including oxygen-dependent degradation (ODD) domain and C-terminal domains are well known to control the stability and activity of HIF-1 α under normoxic conditions; these domains bind with PHD for ODD domain and FIH for C-terminal domain respectively, and then undergo hydroxylation. The importance of N-terminal domain of HIF-1 α has revealed by literatures, however, little is discovered for N-terminal domain-interacting proteins. Therefore, we sought to focus on HIF-1 α N-terminus and, we were very lucky to fish FABP5 as HIF-1 α interacting protein. We hope the review would be satisfied with our explanation.

3. *Fig. 1b. Transfection with an empty FLAG vector should be included as negative control.*

➔ We performed immunoprecipitation with an empty FLAG vector as control in Fig. 3e but omitted to describe it in the figure.

4. *ARNT also binds to the N-terminal region of HIF-1 alpha. The authors should check if ARNT and FABP5 compete for binding to HIF-1alpha.*

➔ Thanks for your comment. As the reviewer mentioned, ARNT also binds to the HIF-1 α N-terminal, and the ARNT/HIF-1 α heterodimerization is required for HIF-1 function in nucleus. However, our result indicated that FABP5 acts as an enhancer for HIF-1 transactivation by inhibiting hydroxylation by FIH. Therefore, binding of ARNT or FABP5 to HIF-1 α occurs separately. FABP5 binds to OA-induced HIF-1 α in the cytosol and FABP5-HIF-1 α complex can't bind to FIH. The stabilized HIF-1 α is translocated into nucleus, and then complex with ARNT and p300 to have full transcriptional activity.

5. *In Fig. 2a overexpressed FLAG-FABP5 is found both in the nucleus and the cytoplasm. In Fig. 4d endogenous FABP5 is only detected in the cytoplasm. Mislocalization could result from overexpression. Experiments with endogenous proteins can help clarify this issue. In addition, proximity ligation assays could be also informative. (same as Rebuttal letter Page 4, Q8)*

➔ This is a very important and critical point. First, we confirmed Fig. 2a and Fig. 4d with three independent experiments. As the reviewer mentioned, overexpressed Flag/SBP-FABP5 is found both in the nucleus and the cytoplasm, while endogenous FABP5 is detected only in the cytoplasm. We think these results are due to limitations of nuclear extraction and western blotting, this prompt us to perform immunocytochemistry, which is more sensitive to detect localization of protein. The results revealed that endogenous FABP5 is found both in the cytoplasm and nucleus (but mostly in cytoplasm), and HIF-1 α co-exists with FABP5 in the cytoplasm and nucleus as indicated yellow color (Fig. 4f). Moreover, we found endogenous HIF-1 α binds to FABP5 and this effect reinforced with OA treated HepG2 cells (Fig. 4g). Although we couldn't afford to conduct proximity ligation assays, we think that FABP5 binds to HIF-1 α in the cytoplasm, thereby inhibiting binding between HIF-1 α and FIH.

6. *Is FABP5 expression regulated under hypoxia and/or by HIF-1?*

→ We think that FABP5 expression is not regulated by oxygen levels nor HIF-1 α based on our data Fig. 2f and 4c. Regardless of oxygen levels, overexpressed FABP5 or endogenous FABP5 were expressed constantly.

7. *The authors should test if the effect of FABP5 on tumor growth is more severe under hypoxia than under normoxia.*

→ We tested the effect of FABP5 on HepG2 cell growth under normoxia and hypoxia with new experiments (Supplementary Fig. 4b). Under hypoxia, cancer cell growth is reinforced compared to normoxia; however, si-FABP5 attenuated HCC cell growth both under normoxia and hypoxia on day 2 and 3.

8. *Fig. 3f and 4g. If FABP5 interacts with the N terminal HIF-1alpha domain, how do the authors explain its effect on Gal4-CAD which only contains the HIF-1alpha CAD? It is possible that FABP5 interacts directly with FIH. The authors should test this in their experiments.*

→ We thank for valuable comment. The Gal4-CAD plasmid was constructed by recombination of CMX-G4 (N) plasmid and HIF-1 α CAD (amino acids 776-826). As the reviewer mentioned, FABP5 might sequester FIH from HIF-1 α CAD by directly interacting with FIH. Multiple effects of FABP5 on HIF-1 α de novo synthesis and transactivation are occurred; however, in this study, we focused the impact of expression of FABP5 on HIF-1 α activity, followed by Fig. 2 to 4. Instead, we performed new experiment to demonstrate binding capability of HIF-1 α to FIH according to the presence of FABP5 in Fig. 4g. We will go to the next research with the direct association of FIH and FABP5. We hope that the reviewer understands our circumstance.

9. *Fig. 3c. A more detailed description of the constructs is needed. It is not clear from the figure or the text, which parts of HIF-1alpha aa sequence are present in each one*

of them. The MW of the GFP inputs should be indicated on the western blot. Empty FLAG vector controls should be included.

→ We added detailed description of the constructs on figure legends (page 31, line 9-11) and the MW of GFP inputs is indicated on western blot (N-terminal ~ 100 kDa; ODDD ~ 120 kDa; C-terminal ~ 70 kDa). Empty Flag vector is not included in the study, since FABP5/HIF-1 α interaction had been already confirmed in Fig. 1b and 3e.

10. *Fig. 3e. The authors should analyze the presence of FABP5 in the immunoprecipitates. In addition, they should show the effect of FABP5 overexpression and FABP5 silencing on the interaction between FIH and endogenous HIF-1 α . (same as Rebuttal letter page 5, Q9)*

→ Thanks for a thoughtful suggestion. We added the blot, which indicates the FABP5 in the immunoprecipitates; IP: HA, IB: Flag (FABP5). In addition, we performed the new experiments that reveal the effect of FABP5 overexpression and FABP5 silencing on the binding between endogenous FIH and HIF-1 α in the Fig. 3f and 4g, respectively. As expected, ectopically expressed FABP5 attenuates the interaction between FIH and HIF-1 α . Moreover, OA-induced FABP5 also decreases the interaction between FIH and HIF-1 α ; however, si-FABP5 treatment reverses this phenomenon.

11. *Fig. 4e and 4f. The increase of HIF-1 α levels shown at Fig. 4e are not in agreement, as should be expected, with a corresponding increase in HIF-1 activity in Fig. 4f. The authors should comment on this discrepancy.*

→ We conducted new experiments on Fig. 4e, and discovered that silencing of FABP5 attenuated HIF-1 α protein levels under both normoxia and hypoxia. This data is grounds for Fig. 4h (now Fig. 4h, Fig. 4f at submission step), and discrepancy was analyzed with three-independent experiments.

12. *Fig. 5c. The sequence of the functional HRE should be shown. The sequence of the primers used for the CHIP should be added to the supplementary figures.*

➔ The functional HRE sequences (CGTG) was stated in figure legends (page 33, line 24-25), and the primer sequences for ChIP assay is added to Supplementary table 4 (Human ACSL1_ChIP- #1 to 3).

13. *The authors should include the following information:*
-The authors should add in the introduction more information regarding the effect of HIFs on lipid metabolism and the effect on cancer cell. A relative review was published recently (Cells 2019, 8, 214; doi:10.3390/cells8030214).

- FABP3, 4 and 7 have been previously shown to be induced by HIF-1 and should be discussed and cited (Cell Rep. 2014, 9, 349–365 and J. Hepatol. 2015, 63, 855–862).

-LIPIN1, CPT1A and ATGL have been previously shown to be regulated by hypoxia and HIFs. The respective papers should be cited (Mylonis et al, J. Cell Sci. 2012, 125, 3485–3493; Liu et al, Toxicol. Lett. 2014, 226, 117–123; Du et al, Nat. Commun. 2017, 8, 1769; Zhang, X. et al. eLife 2017, 6)

➔ Thanks for your thoughtful suggestion. We added the references (Reference number 22, 15, 16, 8, 19, 20, 21, respectively).

14. *Error bars appear to be missing or abnormally small in Figures 3b, 3d (right panel), 4b, 5b, 6c, 6g, 7c and 7f.*

➔ All graphs were replaced with dot plots with error bars and all source data underlying the graphs in the study is available at supplementary information.

We really appreciate these critical comments and eagerly hope that reviewers are satisfied with our revised data and answers.

Chun, Y. S., Choi, E., Kim, G. T., Choi, H., Kim, C. H., Lee, M. J., Kim, M. S., and Park, J. W. (2000). Cadmium blocks hypoxia-inducible factor (HIF)-1-mediated response to hypoxia by stimulating the proteasome-dependent degradation of HIF-1alpha. *Eur J Biochem* 267, 4198-4204.
Furuta, E., Pai, S. K., Zhan, R., Bandyopadhyay, S., Watabe, M., Mo, Y. Y., Hirota, S., Hosobe, S.,

Tsukada, T., Miura, K., Kamada, S., Saito, K., Iizumi, M., Liu, W., Ericsson, J., and Watabe, K. (2008). Fatty acid synthase gene is up-regulated by hypoxia via activation of Akt and sterol regulatory element binding protein-1. *Cancer Res* 68, 1003-1011.

Kuemmerle, N. B., Rysman, E., Lombardo, P. S., Flanagan, A. J., Lipe, B. C., Wells, W. A., Pettus, J. R., Froehlich, H. M., Memoli, V. A., Morganeli, P. M., Swinnen, J. V., Timmerman, L. A., Chaychi, L., Fricano, C. J., Eisenberg, B. L., Coleman, W. B., and Kinlaw, W. B. (2011). Lipoprotein lipase links dietary fat to solid tumor cell proliferation. *Mol Cancer Ther* 10, 427-436.

Lee, G. Y., Shin, S. H., Shin, H. W., Chun, Y. S., and Park, J. W. (2018). NDRG3 lowers the metastatic potential in prostate cancer as a feedback controller of hypoxia-inducible factors. *Exp Mol Med* 50, 1-13.

Mwaikambo, B. R., Yang, C., Chemtob, S., and Hardy, P. (2009). Hypoxia up-regulates CD36 expression and function via hypoxia-inducible factor-1- and phosphatidylinositol 3-kinase-dependent mechanisms. *The Journal of biological chemistry* 284, 26695-26707.

Nieman, K. M., Kenny, H. A., Penicka, C. V., Ladanyi, A., Buell-Gutbrod, R., Zillhardt, M. R., Romero, I. L., Carey, M. S., Mills, G. B., Hotamisligil, G. S., Yamada, S. D., Peter, M. E., Gwin, K., and Lengyel, E. (2011). Adipocytes promote ovarian cancer metastasis and provide energy for rapid tumor growth. *Nat Med* 17, 1498-1503.

Qiu, J. F., Zhang, K. L., Zhang, X. J., Hu, Y. J., Li, P., Shang, C. Z., and Wan, J. B. (2015). Abnormalities in Plasma Phospholipid Fatty Acid Profiles of Patients with Hepatocellular Carcinoma. *Lipids* 50, 977-985.

Sriram, A., Bohlen, J., and Teleman, A. A. (2018). Translation acrobatics: how cancer cells exploit alternate modes of translational initiation. *EMBO Rep* 19.

Reviewers' comments:

Reviewer #1 (Remarks to the Author):

The authors have addressed all of my critiques.

Reviewer #2 (Remarks to the Author):

The authors have made sufficient attempts to address the reviewers queries. Normally I would like to see the in vivo xenograft experiments, but understand that given the current global climate it is difficult. For my part, the manuscript is now suitable for publication.

Reviewer #3 (Remarks to the Author):

In the revised manuscript, the authors have addressed many of the reviewer's questions. However, many important points still need to be clarified.

Specifically:

1. To show the interaction between the endogenous proteins in Fig 4g the authors treat cells with MG132. MG132 is a proteasome inhibitor that affects many cellular pathways. They should demonstrate that the interaction takes place under physiological and relevant to the study conditions i.e. incubation of cells under hypoxia in the presence or absence of OA. Moreover, the increase of the endogenous HIF-1alpha / FABP5 interaction in this figure (figure 4g) is not obvious. In fact, if there is an increase, it is further questionable by the low endogenous FABP5 levels in the absence of OA. The experimental point of FABP5 knockdown in the absence of OA is missing. Quantification and comparison of HIF-1alpha / FABP5 interaction under these conditions could clarify if any difference observed, is merely the result of the different FABP5 expression levels.
2. Appropriate negative controls, such as transfection with an empty FLAG vector, should be included in all experiments. They are important in order to assess both the biological specificity and the technical quality of each experiment.
3. The images in figure 4f raise many questions. The localization of HIF-1alpha in Figure 4f is unexpected, showing equal staining between the nucleus and the cytoplasm. In Figure 4d the LaminB and beta-tubulin controls clearly show contamination between the cytoplasmic and nuclear fractions, yet HIF-1alpha is detected only in the nucleus and FABP5 in the cytoplasm. Therefore, the protein localization shown in Figure 4f could be a result not of OA, but of the MG132 treatment of the cells (see also comment 1). The authors should repeat the experiment under more physiological and relevant to this work conditions (under hypoxia and in the presence or absence of OA) and also do a quantitative analysis of the cytoplasmic and nuclear fractions of the proteins. In addition, the authors should discuss possible scenarios that explain a cytoplasmic localization of OA induced HIF-1alpha. For example, Mylonis et al have shown that ERK 1/2 inactivation leads to increase of HIF-1alpha export from the nucleus and the partial localization of HIF-1alpha in the cytoplasm (Mylonis et al J Biol Chem. 2008 Oct 10;283(41):27620-7) where it interacts on the outer mitochondrial membrane with mortalin (Mylonis et al, J Cell Sci. 2017 Jan 15;130(2):466-479). Could a similar mechanism operate in this case?
4. The authors write "As the reviewer mentioned, FABP5 might sequester FIH from HIF-1α CAD by directly interacting with FIH. Multiple effects of FABP5 on HIF-1α de novo synthesis and transactivation are occurred; however, in this study, we focused the impact of expression of FABP5 on HIF-1α activity, We will go to the next research with the direct association of FIH and FABP5." The authors should add their answer to the discussion and discuss the possibility that

FABP5 also interacts with other parts of HIF-1alpha, as a physical interaction between the two proteins was only shown with full length HIF-1 alpha.

5. Information on the nucleotide sequence surrounding the 4 HRE core nucleotides and their positions in the ASCL1 promoter should also be included in the manuscript.

6. The authors have not discussed, as suggested by the reviewer, previously published findings on the effect of hypoxia on FABPs in references 15 (Bensaad, Cell Rep. 2014, 9, 349–365) and 16 (Hu, J. Hepatol. 2015, 63, 855–862).

Your manuscript entitled "Fatty-acid-induced FABP5/HIF-1 reprograms lipid metabolism and enhances cell proliferation" has now been seen by 3 referees. You will see from their comments below that while they find your work of interest, some important points are raised. We are interested in the possibility of publishing your study in Communications Biology, but would like to consider your response to these concerns in the form of a revised manuscript before we make a final decision on publication.

We therefore invite you to minor revise and resubmit your manuscript in a final round, taking into account the points raised. Please highlight all changes in the manuscript text file.

Collectively, the authors need to address the remaining concerns of Reviewer #3 (e.g. missing negative control).

We are committed to providing a fair and constructive peer-review process. Do not hesitate to contact us if you wish to discuss the revision in more detail or if there are specific requests from the reviewers that you believe are technically impossible or unlikely to yield a meaningful outcome.

Reviewers' comments:

Reviewer #1 (Remarks to the Author):

The authors have addressed all of my critiques.

Reviewer #2 (Remarks to the Author):

The authors have made sufficient attempts to address the reviewers queries. Normally I would like to see the in vivo xenograft experiments, but understand that given the current global climate it is difficult. For my part, the manuscript is now suitable for publication.

Reviewer #3 (Remarks to the Author):

In the revised manuscript, the authors have addressed many of the reviewer's questions. However, many important points still need to be clarified.

Specifically:

1. *To show the interaction between the endogenous proteins in Fig 4g the authors treat cells with MG132. MG132 is a proteasome inhibitor that affects many cellular pathways. They should demonstrate that the interaction takes place under physiological and relevant to the study conditions i.e. incubation of cells under hypoxia in the presence or absence of OA.*

Moreover, the increase of the endogenous HIF-1alpha / FABP5 interaction in this figure (figure 4g) is not obvious. In fact, if there is an increase, it is further questionable by the low endogenous FABP5 levels in the absence of OA. The experimental point of FABP5 knockdown in the absence of OA is missing. Quantification and comparison of HIF-1alpha / FABP5 interaction under these conditions could clarify if any difference observed, is merely the result of the different FABP5 expression levels.

➔ We appreciate the reviewer's comment. HIF-1 α is known to be regulated by hydroxylation; proline 402/564 for oxygen-dependent degradation and asparagine 803 for interruption of co-activator p300/CBP binding, and these phenomena are occurred under oxygen-rich normoxic conditions. As the reviewer mentioned, MG132 is a proteasome inhibitor that affects ubiquitin-proteasome pathways. However, our hypothesis is that FABP5/HIF-1 α binding sequesters FIH from HIF-1 α , we have to grab FABP5/HIF-1 α binding under normoxic conditions, in which FIH activates (our previous report: Li et al., Int. J. Biochem. Cell Biol., 2011 May; 43(5):795-804). For preventing under-estimation of FABP5/HIF-1 α binding by p402/564 hydroxylation dependent degradation of HIF-1 α , HepG2 cells should be treated with MG132 (10 μ M, 8 h). Moreover, we replace the blot (IP: IgG or HIF-1 α , IB: FABP5), and we agreed to the reviewer's comment; Increasing of endogenous FABP5/HIF-1 α binding by OA is merely the result of the different FABP5 and HIF-1 α expression levels.

Because treatment of OA and si-FABP5 regulated input protein level of FABP5 and also HIF-1 α , we thought that FABP5/HIF-1 α interaction was dependent on these two input levels. In Fig.4g, we just wanted to discuss about not only ectopic overexpressed FABP5 but also endogenous FABP5 interacts with HIF-1 α as the reviewer's previous suggestion. We hope the reviewer would be satisfied with our answers.

2. Appropriate negative controls, such as transfection with an empty FLAG vector, should be included in all experiments. They are important in order to assess both the biological specificity and the technical quality of each experiment.

➔ We performed immunoprecipitation in Fig.3e to further confirm Fig. 1b, which showed direct interaction between F/S-FABP5 and HA-HIF-1 α , with an empty FLAG vector. In addition, Fig.3c (HIF-1 α domain study) was replaced with new experiments.

3. The images in figure 4f raise many questions. The localization of HIF-1alpha in Figure 4f is unexpected, showing equal staining between the nucleus and the cytoplasm. In Figure 4d the LaminB and beta-tubulin controls clearly show contamination between the cytoplasmic and nuclear fractions, yet HIF-1alpha is detected only in the nucleus and FABP5 in the cytoplasm. Therefore, the protein localization shown in Figure 4f could be a result not of OA, but of the MG132 treatment of the cells (see also comment 1). The authors should repeat the experiment under more physiological and relevant to this work conditions (under hypoxia and in the presence or absence of OA) and also do a quantitative analysis of the cytoplasmic and nuclear fractions of the proteins.

In addition, the authors should discuss possible scenarios that explain a cytoplasmic localization of OA induced HIF-1alpha. For example, Mylonis et al have shown that ERK 1/2 inactivation leads to increase of HIF-1alpha export from the nucleus and the partial localization of HIF-1alpha in the cytoplasm (Mylonis et al J Biol Chem. 2008 Oct 10;283(41):27620-7) where it interacts on the outer mitochondrial membrane

with mortalin (Mylonis et al, J Cell Sci. 2017 Jan 15;130(2):466-479). Could a similar mechanism operate in this case?

➔ We think the reviewer has raised an important issue. Although our theory is based on the two results; 1) FABP5 binds to HIF-1 α 2) FABP5 interrupts HIF-1 α /FIH interaction. Therefore, interaction between FABP5 and HIF-1 α should be confirmed under FIH activated status, in which oxygen-rich conditions. We designed Fig.4f experiment for visualizing co-localization of FABP5 and HIF-1 α as shown in Fig.4g panel 1 (OA-, si-con) with preventing oxygen-dependent degradation of HIF-1 α (by treatment of MG132 10 μ M for 8 h). As we described in Q1, we firmly believe normoxic MG132 treating conditions are relevant for studying FABP5/HIF-1 α binding, which results in disruption of HIF-1 α /FIH interaction (Li et al., Int. J. Biochem. Cell Biol., 2011 May; 43(5):795-804). In addition, we focused on OA as an inducer for FABP5 expression level. Since ectopically expressed FABP5 enhances HIF-1 α protein level and its transcriptional activity in Fig.2 and 3, we wanted to confirm these phenomena are also occurred via the induced FABP5 by OA treatment. As expected, OA induced the expression FABP5, and this resulted in upregulation of HIF-1 α protein level and transcriptional activity. However, we admit our limitations about a cytoplasmic localization of OA induced HIF-1 α and describe the suggested scenarios in the discussion section (page 15, line: 14-20). Furthermore, we analyzed the cytoplasmic and nuclear fractions of the proteins as the reviewer suggested. We thank the reviewer for the thoughtful comments.

4. The authors write “As the reviewer mentioned, FABP5 might sequester FIH from HIF-1 α CAD by directly interacting with FIH. Multiple effects of FABP5 on HIF-1 α de novo synthesis and transactivation are occurred; however, in this study, we focused the impact of expression of FABP5 on HIF-1 α activity, We will go to the next research with the direct association of FIH and FABP5.’ The authors should add their answer to the discussion and discuss the possibility that FABP5 also interacts with other parts of HIF-1 α , as a physical interaction between the two proteins was only shown with full length HIF-1 α .

➔ We stated this in the discussion part. Please refer to manuscript page 15, line: 3-14.

5. *Information on the nucleotide sequence surrounding the 4 HRE core nucleotides and their positions in the ASCL1 promoter should also be included in the manuscript.*

➔ The 4 HRE core nucleotides (CGTG) and their positions in the ACSL1 promoter; P1 (-2119), P2 (-1491), and P3 (-731) are inserted in the Figure legend on page 34, line 24.

6. *The authors have not discussed, as suggested by the reviewer, previously published findings on the effect of hypoxia on FABPs in references 15 (Bensaad, Cell Rep. 2014, 9, 349–365) and 16 (Hu, J. Hepatol. 2015, 63, 855–862).*

➔ We have inserted these as the reviewer's suggestion in the introduction part (page 3, line 6-7).

We appreciate these comments and hope that reviewers are satisfied with our answers.